

**Multi-Satellite Retrieval of SSA using OMI-MODIS algorithm**
Kruthika Eswaran[1,2*], Sreedharan Krishnakumari Satheesh[1,2] and Jayaraman Srinivasan[1,2]
[1] Centre for Atmospheric and Oceanic Sciences, Indian Institute of Science, Bangalore, India
[2] Divecha Centre for Climate Change, Indian Institute of Science, Bangalore, India
*Correspondence to:* Kruthika Eswaran (kruthika.eswaran89@gmail.com)
**Abstract -** Single scattering albedo (SSA) represents a unique identification of aerosol type and
aerosol radiative forcing. However, SSA retrievals are highly uncertain due cloud contamination
and aerosol composition. Recent improvement in the SSA retrieval algorithm has combined the
superior cloud masking technique of Moderate Resolution Imaging Spectroradiometer (MODIS)
and the better sensitivity of Ozone Monitoring Instrument (OMI) to aerosol absorption. The
combined OMI-MODIS algorithm has been validated over a small spatial and temporal scale
only. The present study validates the algorithm over global oceans for the period 2008-2012. The
geographical heterogeneity in the aerosol type and concentration over the Atlantic Ocean, the
Arabian Sea and the Bay of Bengal was useful to delineate the effect of aerosol type on the
retrieval algorithm. We also noted that OMI overestimates SSA when absorbing aerosols were
present closer to the surface. We attribute this overestimation to data discontinuity in the aerosol
height climatology derived from Cloud-Aerosol Lidar and Infrared Pathfinder Satellite
Observations (CALIPSO) satellite. OMI uses pre-defined aerosol heights over regions where
CALIPSO climatology is not present leading to overestimation of SSA. The importance of
aerosol height was also studied using the Santa Barbara DISORT radiative transfer (SBDART)
model.  The results from the joint retrieval were validated with ground-based measurements and
it was seen that OMI-MODIS SSA retrievals were better constrained than OMI only retrieval.



## 1. Introduction

Aerosols of different types are spatially distributed heterogeneously and at different altitudes in
the atmosphere. Depending upon their properties, certain aerosols (biomass and carbon) warm
the atmosphere by absorbing radiation, while other aerosols (sea salts and sulphates) cool the
atmosphere by scattering radiation (Ramanathan et al., 2001). Due to the opposing effects on the
atmosphere aerosols can have either net warming or cooling effect on the global climate
depending upon the aerosol type, concentration and vertical distribution. Effect of aerosols on the
global climate is measured by 'aerosol radiative forcing' (the perturbation to the earth's radiation
budget caused by the presence of aerosols). Positive forcing implies atmospheric warming and
vice-versa. (Liao and Seinfeld, 1998; Podgorny and Ramanathan, 2001; Satheesh, 2002; Johnson
et al., 2003; Kim et al., 2004; Moorthy et al., 2004; Meloni et al., 2005; Satheesh and Moorthy,
2005; Seinfeld and Pandis, 2006; Satheesh et al., 2008; Chand et al., 2009; Mishra et al., 2015).
According to the climate assessment report, the estimation of aerosol radiative forcing is a major
cause of uncertainty in the estimation of climate sensitivity and therefore presents a great
impediment to climate modeling (IPCC, 2013). The uncertainty is largely due to the lack of
accurate measurement of the scattering and absorbing properties of the aerosols (Cooke and
Wilson, 1996; Menon et al., 2002; Chung and Seinfeld, 2002; Bond and Sun, 2005).
The Single Scattering Albedo (SSA), (the fraction of radiation scattered out of total
extinction of radiation) is used to distinguish the scattering and absorbing properties of aerosols.
SSA represents a unique fingerprint of the type of aerosol and its radiative forcing (Hansen et al.,
1997; Haywood et al., 1997; Myhre et al., 1998). In general, purely scattering aerosols have SSA
value of approximately 1 while highly absorbing aerosols have SSA less than 0.7. However,
SSA values lack high certainty (Bond and Bergstrom, 2006; Bond et al., 2013). Uncertainties in



SSA measurements are due to factors such as cloud contamination, instrumentation error and
aerosol modification due to atmospheric processes. Better SSA retrievals (both in-situ and
satellite-based) are required to reduce the uncertainty in SSA for a more accurate estimation of
aerosol forcing; particularly over regions influenced by a variety of air masses. There is also a
need for accurate spectral aerosol absorption measurements, which is required to validate SSA
derived from satellite measurements (Bergstrom et al., 2007).
Studies on the various direct measurements of SSA and their uncertainty evaluation have
been performed previously (Horvath, 1993, Heintzenberg et al., 1997; Moosmuller et al., 2009).
Along with ground-based retrievals of SSA, there have been other indirect methods to retrieve
the parameter using satellite images and observations (Table 1).
Though these previous studies on ground-based measurements have brought a fundamental
understanding to the estimation of amounts of aerosols / aerosol chemistry, their restricted spatial
and temporal extent is a major limitation. Moreover, these studies also have a reduced
availability of scenes for indirect retrievals. Some techniques are limited due to cloud
contamination while others operate only under specific conditions (e.g. presence of sun glint).
This presents a need for better SSA retrieval algorithms that overcome the present technical
limitations and that can be applied on a global scale. The global extent of observations from
satellites has increased the spatial extent of the observations (Kaufman et al., 2002a). Though the
satellite-based retrievals have been shown to be extremely successful over the majority of ocean
and land regions, they still have a limited success over deserts and ice sheets. Over deserts and
ice-sheets, high surface reflectance affects the satellite retrievals in visible spectrum. To counter
this, SSA is retrieved in UV spectrum (330 nm to 400 nm) over these regions (Torres et al., 1998,
2007). In UV spectrum, the upwelling radiances are highly sensitive to the aerosol absorption



and also have a lower influence of surface albedo (Torres et al., 2007). SSA retrieval in UV
spectrum also avoids difficulties encountered in scenarios where there are large surface
reflectance contrasts.
The quality of OMI SSA retrievals is affected by sub-pixel cloud contamination and the
spectral surface albedo (Torres et al., 2007). To counter the problems and uncertainties in the
OMI SSA retrieval (Table 2), Satheesh et al. 2009 used retrieval from multiple satellites. They
used combined retrieval from OMI-MODIS since sensors on each of the satellites have their own
strengths and both fly within few minutes of each other in the A-train constellation (Stephens et
al., 2002). The better cloud-screened retrieval of AOD from MODIS (Levy et al., 2003) and the
high sensitivity of OMI to aerosol absorption were used to develop a hybrid algorithm to retrieve
SSA (Satheesh et al., 2009). The study was performed over Atlantic Ocean and Arabian Sea for
the year 2006. A comparison of the retrieved aerosol height with aircraft measurements showed
that OMI-MODIS was more accurate than OMI. Gasso and Torres (2016) performed a detailed
analysis of the OMI UV product retrievals over oceans and island sites. They compared the OMI
retrieved AOD with MODIS and AERONET AODs. This work used the OMI-MODIS algorithm
for only two particular cases over and near Africa to understand how the assumption of aerosol
height and shape affected AOD and SSA retrievals. It was found that when the actual height from
satellite Lidar was used instead of climatological values and when the shape of dust aerosols was
assumed to be non-spherical, the retrievals by OMI agreed better with other observations
including OMI-MODIS method. While the OMI-MODIS algorithm has been used in calculating
aerosol radiative forcing (Satheesh et al., 2010) over oceanic regions surrounding India and used
in retrieving SSA over land (Narasimhan and Satheesh, 2013) as well as used to understand the
retrievals of OMI UV products for two particular cases (Gasso and Torres, 2016), a detailed



analysis of the algorithm on a larger spatial and temporal scale has not been done so far.
The current work applies the OMI-MODIS algorithm to retrieve SSA on a global scale. It is
applied over global oceans from 2008-2012. Regional analysis over the Atlantic, the Arabian Sea
and the Bay of Bengal has been done by incorporating the aerosol layer height and the type of
aerosols. A simulation study using Santa Barbara DISORT Radiative Transfer (SBDART) model
was performed to highlight the importance of aerosol layer height. After estimating SSA values
using the OMI-MODIS algorithm, the present study then uses cruise measurements of SSA from
the Integrated Campaign for Aerosols, Gases and Radiation Budget (ICARB) and winter ICARB
campaigns over Arabian Sea and Bay of Bengal in 2006 and 2009 to validate the same (Moorthy
et al., 2008, 2010).
**2.   Data**
**2.1. OMI**
The Ozone Monitoring Instrument (OMI) on board the Aura satellite was launched in 2004. For
OMI measurements two aerosol inversion schemes are used- OMI near UV (OMAERUV)
algorithm and the multi-wavelength (OMAERO) algorithm (Torres et al., 2007). The OMAERO
algorithm uses 19 wavelengths in the range of 330-500 nm to retrieve corresponding aerosol
characteristics. For the present study we have used the OMAERUV algorithm which uses
measurements at two wavelengths 354 nm and 388 nm. The reason behind choosing these
wavelengths is the high sensitivity of upwelling radiances to aerosol absorption and the lower
influence of surface in measurements due to low reflectance values in the UV region. This gives
a unique advantage of retrieving aerosol properties over ocean and land including arid and semi-
arid regions (Torres et al., 1998; 2007).
The  products  derived  from  the  algorithm  include  AOD,  absorption  aerosol  optical  depth



(AAOD) and single scattering albedo (SSA). These are derived from pre-computed reflectance
values for different aerosol models. Three major types of aerosols have been used - Desert dust,
carbonaceous aerosols from biomass burning and sulphate-based aerosols. Each type has seven
models of SSA.  The retrieved products of OMAERUV are sensitive to the aerosol layer height
(Torres et al., 1998). The values are derived at surface and at 1.5, 3.0, 6.0 and 10.0 km above the
surface. The best estimate of the values of AOD, AAOD and SSA of a particular choice of
aerosol vertical distribution are evaluated.

Due to the high sensitivity of SSA retrieval to the assumption of aerosol height and aerosol

type, the OMI algorithm was improved (Collection 003-PGE V1.4.2, Torres et al., 2013) using
climatology of aerosol layer height from CALIPSO (Cloud-Aerosol Lidar and Infrared
Pathfinder Satellite Observations) along with carbon monoxide (CO) measurements from AIRS
(Atmospheric Infrared Sounder) for better identification of carbonaceous aerosols. Torres et al.
(2013) showed that the combined use of AIRS CO measurements and OMI Aerosol Index (AI)
retrievals, helped in identifying the type of absorbing aerosol. Thus smoke layers were identified
when values of AI and CO measurements were high and during events of high AI and low CO
values, the aerosols were identified as dust. The AIRS CO measurements were also used to
identify large aerosol loading which was otherwise represented as clouds by the OMAERUV
algorithm. Using collocated observations of OMI and CALIOP, Torres et al. (2013) estimated the
height of elevated absorbing aerosols for a 30-month period from July 2006 to December 2008.
An effective aerosol height was calculated from the attenuated backscatter weighted with
average height using the CALIOP 1064 nm measurements. The 30-month climatology of aerosol
height was used in the OMAERUV algorithm and validated with Aerosol Robotics Network
(AERONET) observations (Torres et al., 2013). The results showed that there was improvement



in the retrievals. The original aerosol height assumptions were used in the algorithm over regions
where the climatology was unavailable. For the present study we have used the improved
OMAERUV algorithm along with AOD, SSA retrievals at different aerosol heights and as well
as the best estimates of AOD and SSA.
**2.2. MODIS**
The Moderate Resolution Imaging Spectrometer (MODIS) instrument in Aqua satellite was
launched in 2002. This instrument, with 36 spectral channels has a unique ability to retrieve
aerosol properties with better accuracy over both land and ocean (Remer et al., 2005; Levy et al.,
2003). Of these, seven channels (0.47-2.13 μm) are used to retrieve aerosol properties over ocean
(Tanre et al., 1997).
As described in Remer et al., (2005), before the retrieval algorithm, masking of sediments,
clouds and ocean glint is performed to separate valid pixels from bad ones. The retrieval
algorithm of MODIS (also called the inversion procedure) has been described in detail
previously (Tanre et al., 1997; Levy et al., 2003; Remer et al., 2005). The algorithm uses a 'look-
up table' (LUT) approach, i.e., for a set of aerosol and surface parameters, radiative transfer
calculations are performed. Spectral reflectance derived from the LUT is compared with
MODIS-measured spectral reflectance to find the 'best' (least-squares) fit. The resulting
combination of modes provides the aerosol model from which size distribution, properties
including spectral optical depth, effective radius etc. is derived. The product used from MODIS
is the Level 2 aerosol (MYD04, Collection 5.1) product. The parameter chosen is
'Effective_Optical_Depth_Average_Ocean' which provides the aerosol optical depth over ocean
at seven wavelengths. The value is the average of all the solutions in the inversion procedure
with the least-square error $< 3\%$.





A combination of OMI and MODIS helps indirectly in counteracting the cloud
contamination problem and also uses the strength of the individual sensors – OMI's sensitivity to
aerosol absorption combined with the better cloud screening of MODIS and accurate retrieval of
AOD, and aerosol size (Satheesh et al., 2009; Narasimhan and Satheesh, 2013).
**3.    Algorithm**
MODIS has high spatial pixel resolution of 10km x 10km at nadir (and a cloud mask at 500m
and 1km resolution) whereas OMI has a resolution of 13 km x 24 km. This results in a pixel
being prone to cloud contamination which overestimates AOD and underestimates single
scattering co-albedo (1-SSA) (Torres et al., 1998). However, AAOD can be retrieved in the
presence of small cloud contamination since there is cancellation of errors (Torres et al., 2007).
The higher accuracy in MODIS retrieval over ocean is due to the fact that it has large
number of channels in the Shortwave Infrared (SWIR) region (Tanre et al., 1997; Remer et al.,
2005; Levy et al., 2003). While OMI is highly sensitive to aerosol absorption in the near-UV
region, the accuracy in the retrieval of AAOD depends on the aerosol layer height assumption.
OMI provides AOD and AAOD at different heights as prescribed by various aerosol types
(Torres et al., 2007).
The assumption of aerosol layer height in the OMI algorithm restricts the retrieval of AOD
and AAOD. Using this as basis, the approach proposed in Satheesh et al. (2009) used MODIS
AOD as an input to the OMI retrieval algorithm, so that the inversion, now checked, can use the
information to infer the aerosol layer height and SSA. To know the SSA at 388 nm, the AOD
used should also be at the same wavelength. Satheesh et al. (2009) extrapolated MODIS AOD
and compared the estimated UV AOD with high quality ground-based AERONET observations.
The deviation between MODIS-extrapolated AOD and AERONET AOD was greater at higher





AERONET AOD values. This was attributed to the presence of large number of fine-mode
aerosols which affected AOD at UV wavelengths. Hence to improve the linear extrapolation,
information on the aerosol spectral curvature was also included. This was achieved by using an
average regression equation to correct the MODIS AOD (Satheesh et al., 2009; Equation 3).
They showed that MODIS AOD can be linearly extrapolated to 388 nm and use the corrected
AOD as input to the OMI retrieval algorithm. The present work uses the same algorithm as
proposed by Satheesh et al. (2009) to retrieve SSA over the oceans for the region 60S-60N and
180W-180E from December 2007-November 2012. The methodology is described in detail in the
following section.
**4. Methodology**
The AOD for ocean obtained from the Level 2 aerosol product of MODIS (MYD04) was used.
Using linear extrapolation, AOD at 388 nm (hereafter, $AOD_{388}$) was calculated from AOD at
seven wavelengths ranging from 0.47-2.13 μm, after the inclusion of aerosol spectral curvature
defined in Satheesh et al. (2009). OMI provides AOD and SSA for five different aerosol layer
heights starting from surface and at 1.5, 3.0, 6.0 and 10.0km ($AOD_{omi}$ and $SSA_{388}$). It also
provides the best estimate of SSA calculated for a particular aerosol vertical distribution
($SSA_{omi}$).
For the present study, polar regions are not included and hence pixels from both OMI and
MODIS that are outside the 60S-60N and 180W-180E region are excluded. Pixels with invalid or
missing values are also excluded. To reduce computation time the various parameters extracted
from the data were re-gridded onto a uniform grid of 0.5˚ x 0.5˚ within the region of study. For
both the satellites, this procedure was repeated for each swath data which were then combined to
calculate the daily means.





The daily data from collocated MODIS and OMI were utilised in the final algorithm. As
mentioned before OMI provides AOD and SSA for five different aerosol layer heights. Using
$AOD_{388}$ as the reference, the corresponding aerosol layer height was calculated from the five
$AOD_{omi}$ values through linear interpolation. This height is then used as a reference to find the
SSA using interpolation from the set of $SSA_{388}$ values. Finally, this SSA ($SSA_{omi-modis}$), and the
best estimate of SSA ($SSA_{omi}$) were compared to each other.
**5.    Results**
The spatial distribution of SSA retrieved using OMI is shown in Fig. 1a. The values are averaged
over five years and plotted seasonally.
The SSA retrieved using OMI-MODIS algorithm is shown in Fig. 1b.
SSA over open oceans is close to 1 due to the presence of large amount of sea-salt and
sulphate. Closer to land, a variety of aerosols are present which results in SSA varying from 0.75
to ~1. Over the oceans, separating ocean colour effects and aerosol concentrations is difficult.
Hence the OMI algorithm retrieves only if there are enough absorbing aerosols present, i.e.AI
>=0.8 (Torres et al., 2013). Only pixels whose quality has been assigned as 0 or the highest
quality by OMI have been used. Since 2007, observations have been affected by a phenomenon
called the *row anomaly* which reduces the quality of radiance at all wavelengths. The points
flagged for row anomaly are not used in this study. Further information about row anomaly can
be found in Jethva et al. (2014). Thus, the retrievals did not cover the entire globe. From Fig.1a it
can be seen that majority of the valid SSA retrievals were over major aerosol sources in the
world and not over remote oceanic regions like central equatorial Pacific or Antarctic ocean. The
major sources include the vast biomass outflow over Atlantic Ocean from the west coast of
Africa, the dust over Arabian Sea from the arid areas of Arabia & Africa and the dust blown over



Atlantic Ocean from Sahara. Other regions like the east coast of China, Bay of Bengal are
influenced by a variety of anthropogenic aerosols during different seasons. Both the algorithms
capture the major oceanic regions which are influenced by large number of aerosols.

Two important regions over oceans influenced by a variety of aerosols are the Atlantic

Ocean and the oceans around the Indian subcontinent.  The new approach was used over these
regions- Atlantic (5N-30N; 60W-20W) (ATL) and Arabian Sea and Bay of Bengal (0-25N; 55E-
100E) (ARBOB).
**5.1. Difference in SSA retrieval algorithms during different seasons**
To understand how the OMI-MODIS algorithm compared with the retrieval using existing OMI
algorithm, the difference between $SSA_{omi-modis}$ and $SSA_{omi}$ ($\Delta SSA$) averaged over five years for
different seasons is shown in Fig. 2.

During March-April May (MAM) and June-July-August (JJA), there is a longitudinal

gradient in $\Delta SSA$ from the coast of Sahara towards the open Atlantic Ocean. Kaufman et al.
(2002a) showed that closer to the coast of Africa, aerosols are more absorbing than those away
from the coast. The difference in the type of aerosols as we move away from the coast could be
one of the reasons for the gradient in $\Delta SSA$. The $\Delta SSA$ changes sign with season. This was
attributed to the dominating presence of either natural aerosols (JJA) or anthropogenic aerosols
(DJF).

Both ATL and ARBOB regions are influenced by the type of aerosols which result in a

complex mixture and eventually resulting in the variation in SSA distribution over each season.
While the spatial plot of $\Delta SSA$ in Fig. 2 represents the regions where maximum and minimum
differences are located around the globe, a distribution plot provides the ranges of $\Delta SSA$ which
dominate and which do not. The distribution of $\Delta SSA$ for different seasons averaged over five



years (2008-2012) is plotted in Fig. 3a and 3b for the regions- ATL and ARBOB respectively.
DJF shows a strong positive bias in both the regions, JJA shows a negative bias and the
other two seasons show negligible bias. While dust outflows dominate over ATL, over ARBOB –
Arabian Sea is affected by dust at higher altitudes and sea-salt near the surface whereas the Bay
of Bengal is influenced mainly by continental and marine aerosols. The change in the sign of
difference could either be due to the difference in type of aerosol or the assumption in aerosol
layer height (ALH). To understand what type of aerosols affect these water bodies, trajectory
analysis is done. This helps in identifying major sources of aerosols during each season.
**5.2. Trajectory analysis**
**5.2.1.  Atlantic (ATL)**
The region in the tropical Atlantic is surrounded by the Sahara Desert in the east and the
North America in the west. The transport of dust from Sahara over Atlantic Ocean is a regular
occurrence (Prospero and Carlson, 1972). Aerosol distribution over Atlantic is also affected by
the African Easterly Waves and other atmospheric dynamics in Africa (Zuluaga, 2012). The
Atlantic region is influenced by not only dust from Sahara, but also by aerosols from biomass
burning off the coast of Africa and aerosols from industries and pollution from America. Thus,
there is a complex mixture of aerosols over the Atlantic Ocean during any season. A 7-day back
trajectory analysis was performed at a location in the box (15N; 45W) using the online Hybrid
Single-Particle Lagrangian Integrated Trajectory (HYSPLIT) model for the years 2009-2010.
The trajectory was computed for different seasons at 3 heights – 500m, 1500m and 2500m above
mean sea level (MSL). The Atlantic Ocean was divided into four quadrants representing the
regions of possible sources of aerosols 1) North America, 2) Central/South America, 3) North
Africa and 4) Southern Africa (Fig. 4). The influence of these aerosol sources over Atlantic




Ocean is estimated as the percentage of trajectories that start from each region respectively. The
maximum influence is given in bold (Table 2).

From Table 2 it can be seen that the major source of aerosols over the Atlantic Ocean is the

dust outflow from the Sahara Desert (Prospero, 1996). Extreme heating over Sahara creates a
layer of instability (Saharan Air Layer) which lifts the dust particles enabling long-range
transport. Far off the coast the warm dust layer encounters a cooler, wetter air layer causing
inversion. This results in the dust layer being intact over Atlantic Ocean (Prospero and Carlson,
1972). Field experiments like the trans-Atlantic Aerosol and Ocean Science Expeditions
(AEROSE I and II) showed the outflow of dust during spring and summer along with other trace
gases and biomass aerosols (Morris et al., 2006). However, dust is not the only aerosol present in
the region of study. Using an airborne differential absorption LIDAR (DIAL) system, Harriss et
al. (1984), found that there is advection of anthropogenic pollutants from North America to the
North Atlantic Ocean. Advanced very high-resolution radiometer (AVHRR) instrument on the
National Oceanic and Atmospheric Administration (NOAA) 11 satellite provides global aerosol
information. From that data it was found that large plumes over Atlantic Ocean were attributed to
the pollution from North America and Europe. During spring and summer, the large outflow was
due to the dust outbreak from Sahara and Sahel. Biomass burning from southern Africa, South
America and anthropogenic emissions from North and Central America dominated the aerosol
loading over Atlantic Ocean during winter (Husar et al., 1997). The MODIS instrument onboard
the Terra satellite was first used to study the transport and deposition over Atlantic Ocean. It was
found that during winter, the dust which was present was mixed with the biomass aerosols from
Sahel and closer to the coast of North America the dust was influenced by the pollution and
smoke from the continent. Pure dust was present over the ocean during summer months





(Kaufman et al., 2005). From Table 2 it is also seen that the dust dominated at all heights except
during winter when the pollution from North America dominated at higher altitudes.
**5.2.2.  Arabian Sea and Bay of Bengal (ARBOB)**

The Arabian Sea and the Bay of Bengal are oceanic regions on the west and east coast of

India respectively. Both regions are influenced by various types of aerosols during different
seasons. The Arabian Sea has been dominated by dust aerosols and is influenced by high levels
of dust during certain seasons as seen from satellite images (Sirocko and Sarnthein, 1989). Pease
et al. (1998) studied the geochemistry and the transport of various dust samples during different
cruises in different seasons. During winter and summer, the pattern of aerosol transport was
similar to that of the Indian monsoon pattern – northeasterly (winter) and southwesterly
(summer). Thus, the major sources of aerosols were the Arabian Peninsula (including Saharan
dust and Middle East) and Indian sub-continent in summer and winter respectively. The mean 7-
day back trajectory using HYSPLIT model from a point over Arabian Sea (15N; 65E) was
performed for each season of 2010 and at three different heights (500m, 1500m and 2500m
above MSL). Only one year is performed since the trajectory analysis over Atlantic Ocean
showed that the aerosol pathways did not vary much between years. The Arabian Sea region was
divided into four quadrants – 1) Arabian Peninsula and North Africa, 2) Southern Africa, 3)
Indian sub-continent and 4) Indian Ocean and Southeast Asia (Fig. 5). Similar to Table 2,
influence of different aerosol source regions over the Arabian Sea is given in Table 3.

Similar to Pease et al. (1998), Tindale and Pease (1999) found that transport of aerosols near

the surface followed the surface wind currents. The dust content was low near the surface during
summer due to the presence of Findlater jet, but the general dust concentrations were higher than
other oceanic regions. During winter, the winds are predominantly north and north easterly and



hence results in transport of aerosols from India/Pakistan/Afghanistan onto Arabian Sea.
However, the presence of anticyclonic circulation over Arabia (20N; 60E) results in north
westerly winds transporting dust over Arabian Sea (Rajeev et al., 2000). The spring time (March-
April-May) is the transition between northeast and southwest monsoon. The winds become south
westerlies which result in the advection of aerosols from open Indian Ocean or near Somalia. At
higher altitudes (above the Findlater jet) dust transport occurs from Arabia. During summer, the
southwest monsoon wind patterns carry aerosols all the way from southeast/east Indian Ocean
(mainly sea-salt). As the altitude increases, the wind patterns change a little due to aerosols
coming from southwest Indian Ocean/Somalia. Above the Findlater jet, as explained by Tindale
and Pease (1999), dust transport occurs from Arabian Peninsula (Table 3).
Being an integral part in the Indian Summer Monsoon, studies over Bay of Bengal is
important especially the role of aerosols in the local climate change. While Arabian Sea is
dominated by dust and oceanic aerosols and only anthropogenic aerosols during SON, studies
have shown that Bay of Bengal is influenced by various air masses associated with Asian
monsoon system (Krishnamurti et al., 1998). The synoptic meteorological conditions over Bay of
Bengal have been studied in detail by Moorthy et al. (2003) and Satheesh et al. (2006). Similar to
the other two regions, mean 7-day back trajectory analysis from a point over (15N; 90N) was
performed for each season of 2010 and at three different heights (500m, 1500m and 2500m
above MSL). The four quadrants representing the various aerosol source regions are 1)
India/Arabian Peninsula, 2) Indian Ocean, 3) North/Northeast India and East Asia and 4)
Southeast Asia (Fig. 6). Table 4 represents the influence of aerosol source regions over Bay of
Bengal.
The north westerly winds occur from west to east in the Indo-Gangetic Plain (IGP) and due



to subsidence, the aerosols are trapped in the east during winter (Dey and Di Girolamo, 2010; Di
Girolamo et al., 2004). The IGP with its heavy population and large number of industries acts as
a source for anthropogenic aerosols which are transported to Bay of Bengal during winter
(Kumar et al., 2013). Along with mineral dust from Arabian Peninsula, biomass aerosols from
Southeast Asia are also transported to the bay. Field experiments like ICARB (Moorthy et al.,
2008) during the spring time (pre-monsoon) showed transports of aerosols from the Arabian
Peninsula and also presence of elevated aerosols (anthropogenic and natural) over Bay of Bengal
(Satheesh et al., 2008). The post monsoon season acts as a transition from the summer to winter
monsoon. The winds during September are still south westerlies and during October weak
westerlies are present (Lawrence and Lelieveld, 2010). This results in transportation of aerosols
from Indian Ocean and Arabian Sea. Thus, from Table 4 it can be seen that both anthropogenic
aerosols (from IGP, Southeast Asia) and natural aerosols (marine and dust) are present over Bay
of Bengal during different seasons.
**5.3. Role of Aerosol Layer Height in SSA retrieval**
Satheesh et al. (2009) devised a new algorithm to improve the retrieval of SSA using
combined OMI and MODIS data. They used MODIS-predicted UV AOD as the input to improve
the original OMI algorithm, which was constrained by the assumption of aerosol layer height.
Over the Atlantic, the values retrieved from both algorithms showed reasonably good agreement.
However, over the Arabian Sea only when there was considerable loading of dust, the OMI AOD
and MODIS AOD had agreement suggesting that during other seasons, the assumption of aerosol
height could be wrong. Satheesh et al. (2009) also found that over Arabian Sea the aerosol layer
height (ALH) derived from OMI-MODIS algorithm agreed well with aircraft measurements
when compared to OMI SSA retrieval. In the current work, the aerosol layer height (ALH) was



calculated for OMI, using the best estimate of SSA retrieved from OMI. The difference in
aerosol layer height between OMI-MODIS and OMI was plotted with the difference in SSA (Fig.
7a and 7b). The colorbar in the figure represents height estimated using the OMI-MODIS
algorithm.
Most important observation from this analysis is that OMI overestimates SSA at lower ALH
(retrieved by OMI-MODIS algorithm) and underestimates SSA at higher ALH. The latest version
of OMI algorithm uses CALIPSO climatology of aerosol layer height for better accuracy.
However, over regions where this is not available, pre-defined aerosol height has been used
based on the type of aerosol assumed. For industrial sulphate aerosols exponential profile with
2km scale height is assumed with a similar profile with 1.5km scale height for oceanic aerosols.
For biomass type aerosols, a Gaussian distribution with peak at 3km is used. Dust aerosols are
assumed to have two-single Gaussian distributions with maximum at heights 3 and 5km. It has
been shown by Gasso and Torres (2016) that when the actual aerosol height was 1.5km more
than climatological or assumed height, OMI retrieved higher SSA.
It can be seen from Figs. 7a and 7b, the blue coloured circles represent height between
surface to ~ 2km. In this range it is seen that the height assumed by OMI is > 1.5km compared to
the one estimated by OMI-MODIS. Thus, OMI overestimates SSA compared to the OMI-
MODIS retrieval. This overestimation is due to the predefined vertical profiles. Thus, there are
errors with regard to both the aerosol layer height as well as the type of aerosol in the OMI
algorithm. In the OMI algorithm, the highest uncertainty in retrieving SSA is due to aerosol layer
height and aerosol type (Torres et al., 2002). Using ground-based LIDAR measurements,
Satheesh et al. (2009) concluded that OMI-MODIS retrieved height agreed better with
observations than OMI.



The importance of ALH and SSA in the calculation of TOA flux is studied using Santa
Barbara DISORT (SBDART) model (Ricchiazzi et al., 1998). For the same tropical environment
variables and surface albedo of 0.06, the SSA was varied from 0.8 to 1 and aerosol height from 0
to 10 km at 1 km interval. The simulations were done for a narrow band in UV (300-400nm). For
a constant AOD, AE (Angstrom Exponent) and asymmetry factor (0.4, 1 and 0.7 respectively),
TOA flux was calculated (Fig. 8). It can be seen that at any ALH, TOA flux varied with SSA in.
The role of ALH is important in the UV region due to the phenomena of Rayleigh scattering (van
de Hulst, 1981). The importance of Rayleigh scattering on the role of ALH is further shown in
Fig. 9. In this particular set of simulations, the Rayleigh scattering is completely removed and all
other parameters are kept same as in Fig. 8.
It can be seen that once molecular scattering is removed, the effect of ALH is also removed
and TOA flux depends only on SSA and other aerosol properties. This set of SBDART
simulations shows us how for a particular value of TOA flux, assuming different aerosol height
gives us different SSA values reiterating the important role of aerosol height on SSA retrievals.
**5.4. Validation**
To validate the new retrieval method of SSA using OMI and MODIS, both SSA values from
OMI and OMI-MODIS were compared with ground-based measurements (SSA at 450nm)
during Cruises in the period 2006 and 2009 in Arabian Sea and Bay of Bengal. These cruises
were part of the Integrated Campaign for Aerosols, gases and Radiation Budget (ICARB)
performed during the months of March to May 2006 and once during winter (W-ICARB) from
27 December 2008 to 30 January 2009 (Moorthy et al., 2008 and 2010). Since the spatial
coverage of OMI-MODIS and cruise measurements is less, the SSA values for both the
algorithms were averaged over the region of study and compared with observed SSA (Fig. 10).





However, the cruise measurements showed that SSA varied a lot spatially especially over Bay of
Bengal. Hence instead of a spatial average, the SSA values were temporally averaged for the
months when the cruise was performed. This was done under the assumption that during the
cruise period, the SSA over each location did not vary with time. For better coverage, a 1.5° box
was used around each location within which the mean SSA was calculated.
The mean SSA of OMI, OMI-MODIS and cruise measurements are calculated and the
difference between mean satellite SSA and mean SSA from cruise measurements are calculated
for OMI and OMI-MODIS algorithms separately. A statistical t-test is performed comparing the
respective SSA means of OMI and OMI-MODIS with SSA. The null hypothesis assumes the
mean SSA of OMI/OMI-MODIS is equal to the mean SSA calculated from the cruise
measurements. The values from Table 5 show that despite the mean difference of OMI SSA and
cruise SSA being ~ 0.013, it was statistically significant at 95% significance level. On the other
hand the SSA retrieved using OMI-MODIS algorithm was better constrained and was closer to
the mean value of SSA from cruise measurements.  The distribution of SSA from both the
satellite algorithms as well as from cruise measurements is shown in Fig. 11.
Using five years (2008-2012) of OMI and OMI-MODIS data for the region of Arabian Sea
and Bay of Bengal, SSA was retrieved and the difference between the two methods was
calculated and plotted against SSA from the OMI-MODIS algorithm (Fig. 12). For absorbing
aerosols detected by OMI-MODIS the SSA is overestimated by OMI.
The OMI-MODIS approach in SSA retrieval is one of the many combinations of sensors that
can be used in retrieving aerosol properties. A more complete approach involving better vertical
distribution of aerosols either from space or ground-based observations is required to reduce the
uncertainty further. However, with few ground-based measurements in the UV regime, validation





of new algorithms is still in the nascent stage.
**6.    Summary and Conclusions**

Aerosol forcing depends on aerosol properties like aerosol optical depth (AOD) and single

scattering albedo (SSA). SSA is highly sensitive to the aerosol composition and size and as well
as the wavelength at which the aerosol interacts with radiation. A slight change in SSA value can
alter the sign of the forcing. Hence it is important to have an accurate measurement of SSA
globally. Ozone Monitoring Instrument (OMI) retrieves SSA in the UV spectrum. However,
these retrievals are affected by cloud contamination and are sensitive to aerosol layer height. To
resolve the issue of sub-pixel cloud contamination, Satheesh et al (2009) developed a method
using the combination of OMI and the Moderate Resolution Imaging Spectroradiometer
(MODIS) at a local scale. In the present study, we use the method developed by Satheesh et al
(2009) to retrieve SSA at a much larger spatial and temporal scale. The main findings from our
study are listed below:

1. Both OMI and OMI-MODIS algorithms retrieved SSA over regions influenced by large

amounts of aerosols (e.g. Atlantic Ocean – ATL; Arabian Sea and Bay of Bengal –

ARBOB)

2. Difference in SSA retrievals of OMI-MODIS and OMI for both regions ATL and

ARBOB fluctuates between positive and negative values during different seasons which

could be due to the difference in either the type of aerosol or aerosol height assumed. In

addition, a longitudinal gradient of difference in SSA retrievals is present from the coast

of Sahara to the open ocean during the JJA season. This could be due the difference in

type of aerosols near the coast and in the open ocean

3. OMI overestimates SSA at lower ALH and underestimates at higher values of ALH. Over



regions where CALIPSO climatology is not present, OMI uses pre-defined aerosol
heights based on the aerosol present. From Fig. 4 it is also seen that OMI is unable to
retrieve absorbing aerosols present at very low heights (< 2km) due to the already defined
vertical profiles.
4. In the UV spectrum, ALH plays a more dominant role than in the visible region due to
the major effect of Rayleigh scattering in UV. When Rayleigh scattering was removed,
ALH had no effect in both the UV and visible regions of the spectrum.
5. OMI-MODIS method was validated using cruise data from the ICARB and W-ICARB
campaigns in the Arabian Sea and Bay of Bengal. The difference between OMI SSA and
SSA from cruise measurements despite being small is statistically significant. OMI-
MODIS SSA is better constrained and is closer to the cruise measurements
6. It is seen that the OMI overestimates SSA when absorbing aerosols were detected by
OMI-MODIS and the cruise measurements.
Aerosol type and aerosol layer height play a very important role in the retrieval of aerosol
properties. Without the assumption of aerosol type or height, OMI-MODIS is able to detect
absorbing aerosols much better than OMI. Hence this algorithm is useful over regions dominated
by absorbing aerosols like Bay of Bengal during winter. The importance of aerosol height is
clearly demonstrated by SBDART model and the validation with ground-based measurements
highlighted the role of aerosol type. However, an accurate comparison and validation of such
retrieval algorithms can be possible only when there are more ground-based observations
available in the UV spectrum on a larger spatial and temporal scale.
**Acknowledgements**
The authors gratefully acknowledge the NOAA Air Resources Laboratory (ARL) for the



provision of the HYSPLIT transport and dispersion model used in this publication. The authors
are grateful to NASA data and services centre.

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





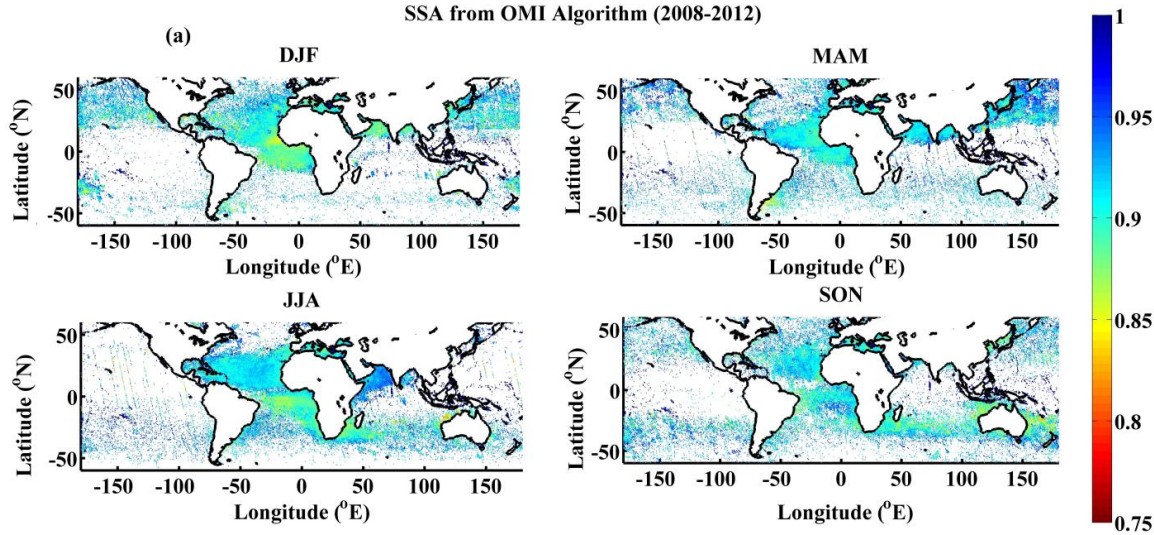


**Figure 1a.** Spatial distribution of SSA retrieved by OMI

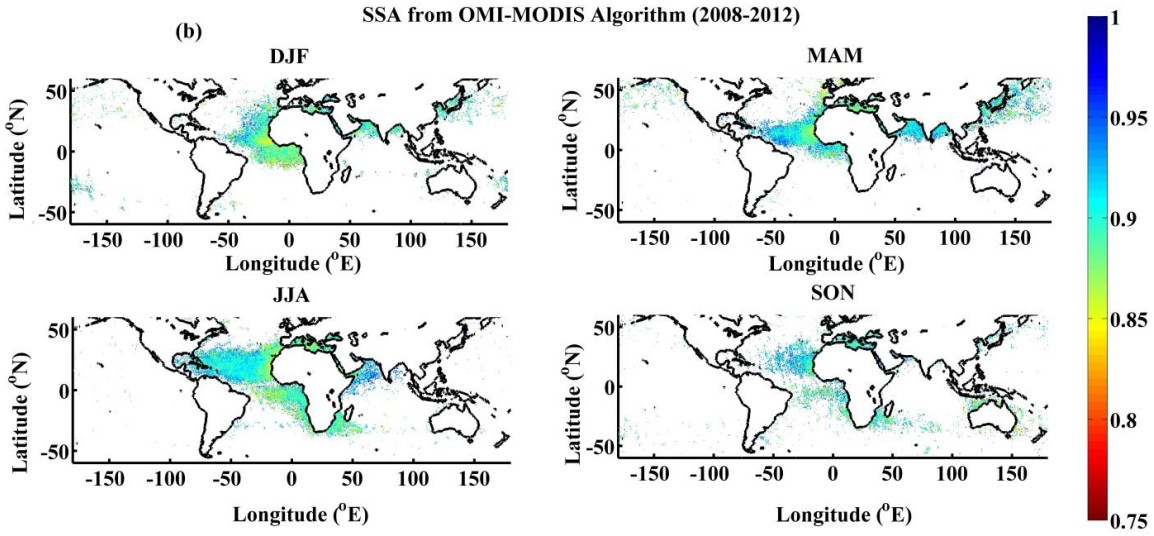


**Figure 1b.** Spatial distribution of SSA retrieved by OMI-MODIS



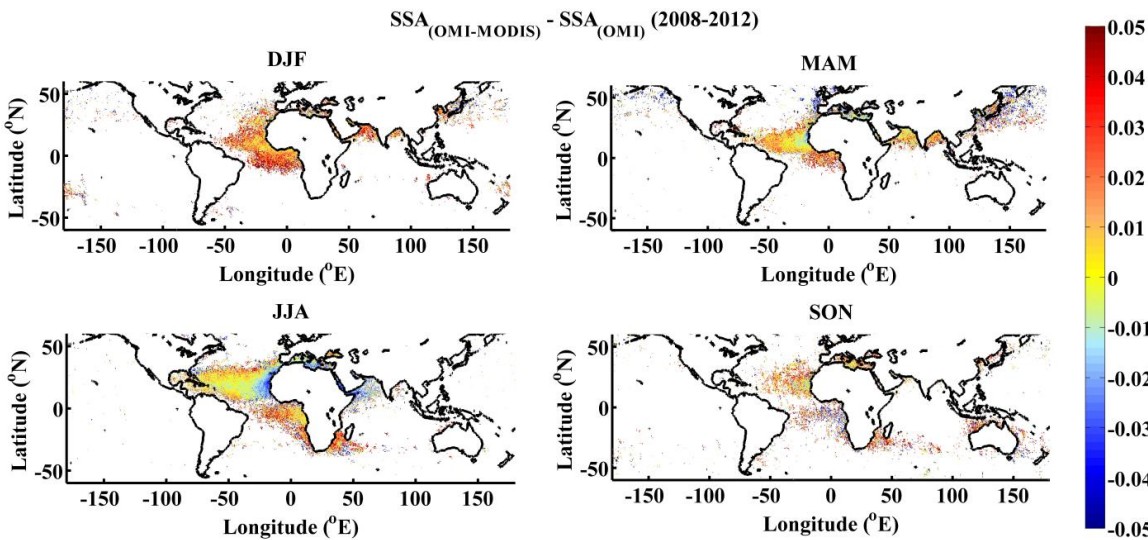


**Figure 2.** Spatial distribution of difference in SSA retrievals










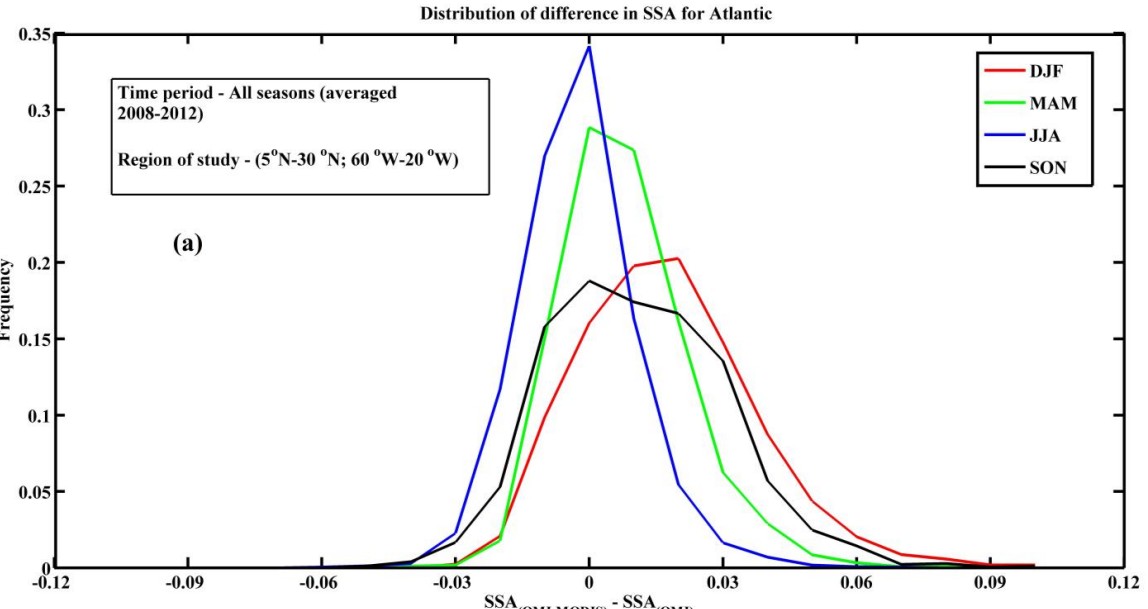


**Figure 3.** Distribution of difference in SSA for all seasons averaged over 2008-2012 over a)

Atlantic and b) Arabian Sea and Bay of Bengal






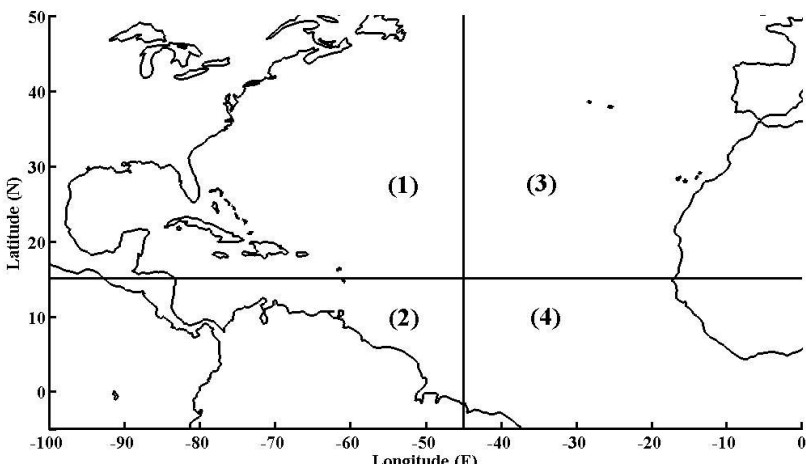


**Figure 4.** Regions representing the various aerosol sources over Atlantic Ocean. 1) North

America, 2) Central/South America, 3) North Africa and 4) Southern Africa.

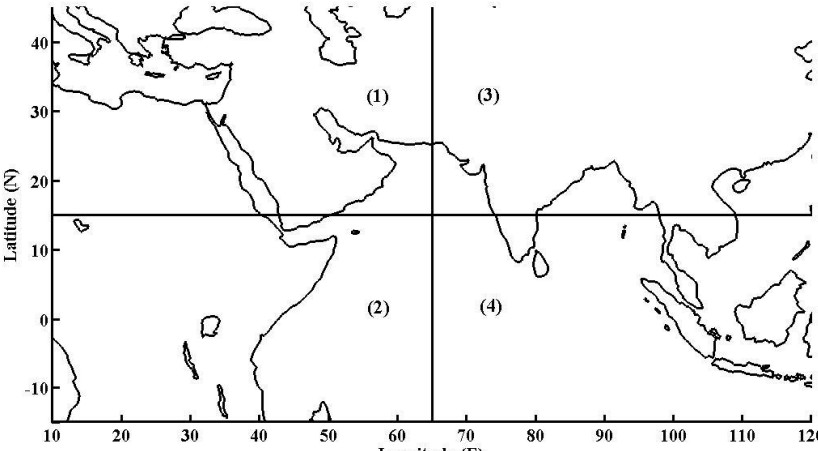


**Figure 5.** Regions representing the various aerosol sources over Arabian Sea. 1) Arabian

Peninsula and North Africa, 2) Southern Africa, 3) Indian sub-continent and 4) Indian Ocean and

Southeast Asia.



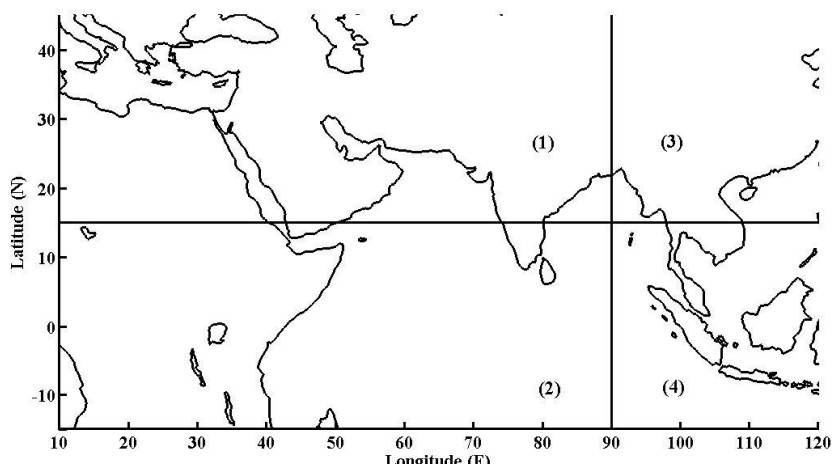


**Figure 6.** Regions representing the various aerosol sources over Bay of Bengal. 1) India/Arabian
Peninsula, 2) Indian Ocean, 3) North/Northeast India and East Asia and 4) Southeast Asia.

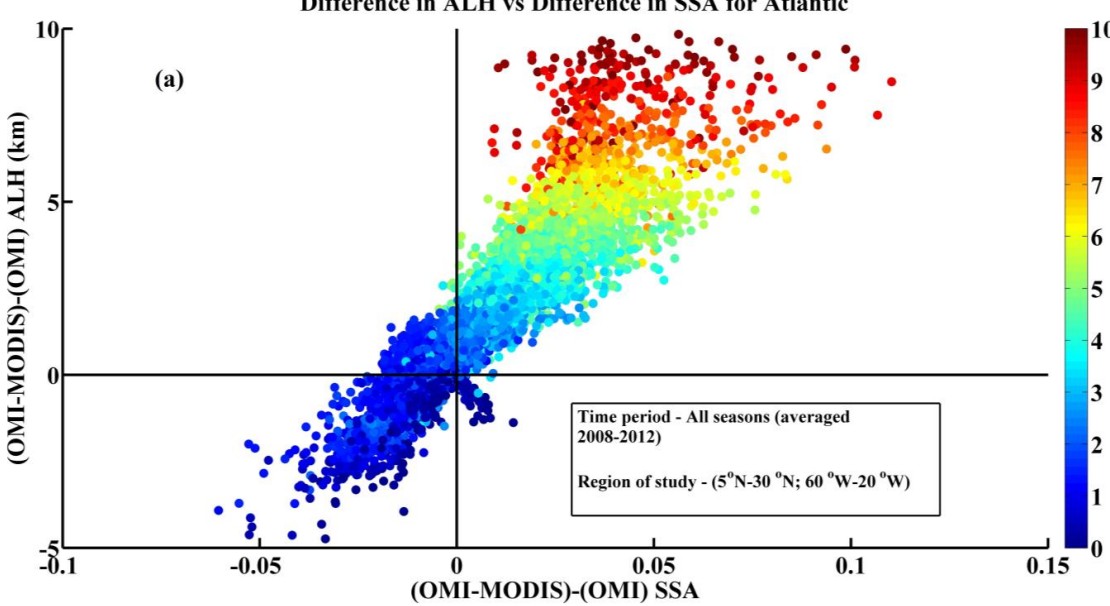




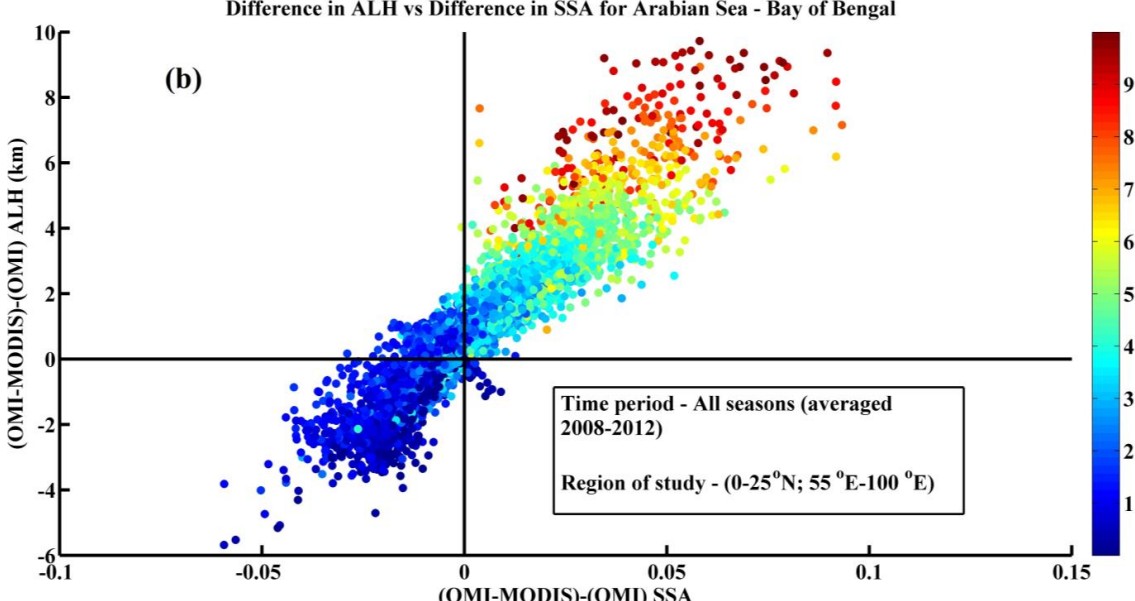


**Figure 7.** Difference in aerosol layer height (ALH) between OMI-MODIS and OMI vs.

difference in SSA over a) Atlantic and b) Arabian Sea and Bay of Bengal. The colorbar

represents ALH estimated by OMI-MODIS algorithm. At lower height (dark blue circles) OMI

assumes ALH greater than that of OMI-MODIS and results in overestimation of SSA.

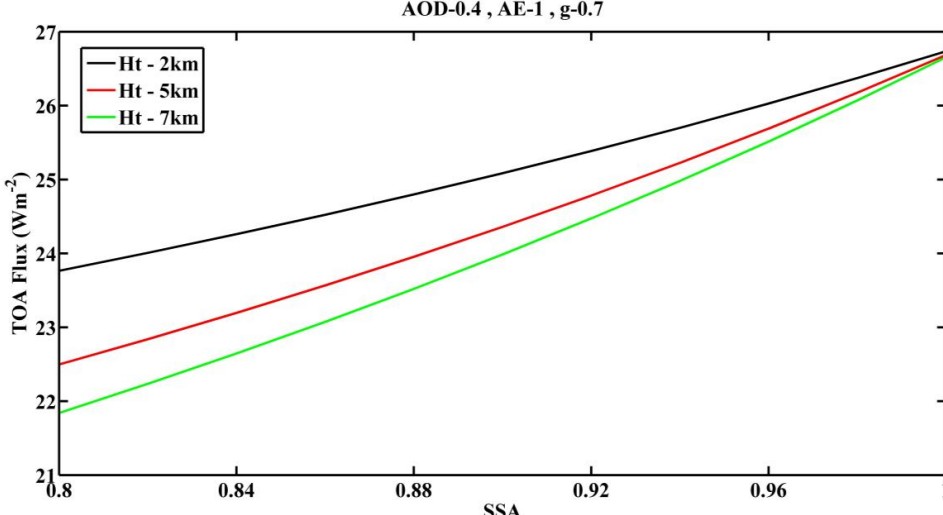


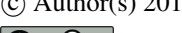



**Figure 8.** TOA flux calculated from SBDART for different SSA and ALH for UV (300-400nm)

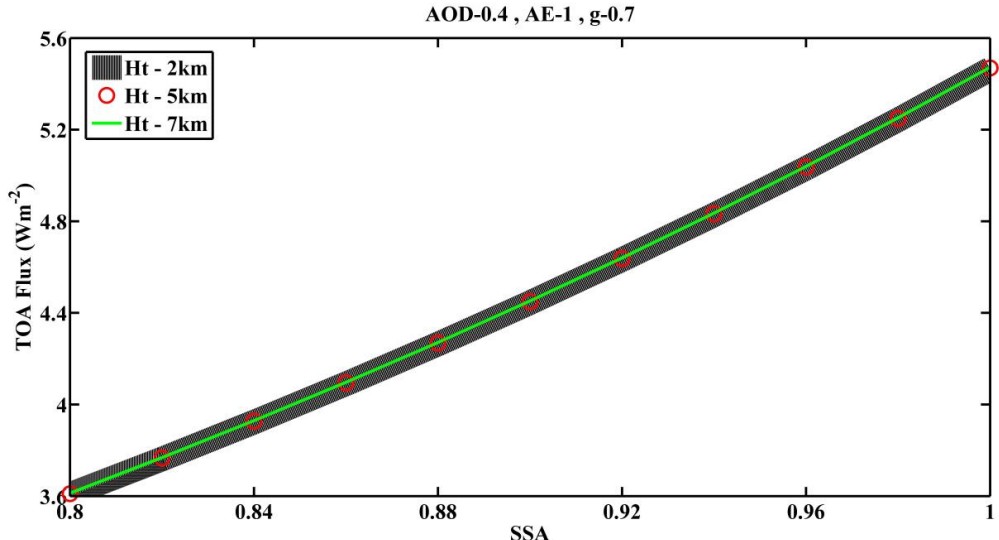


**Figure 9.** TOA flux calculated from SBDART for different SSA and ALH with Rayleigh
scattering removed for UV (300-400nm)

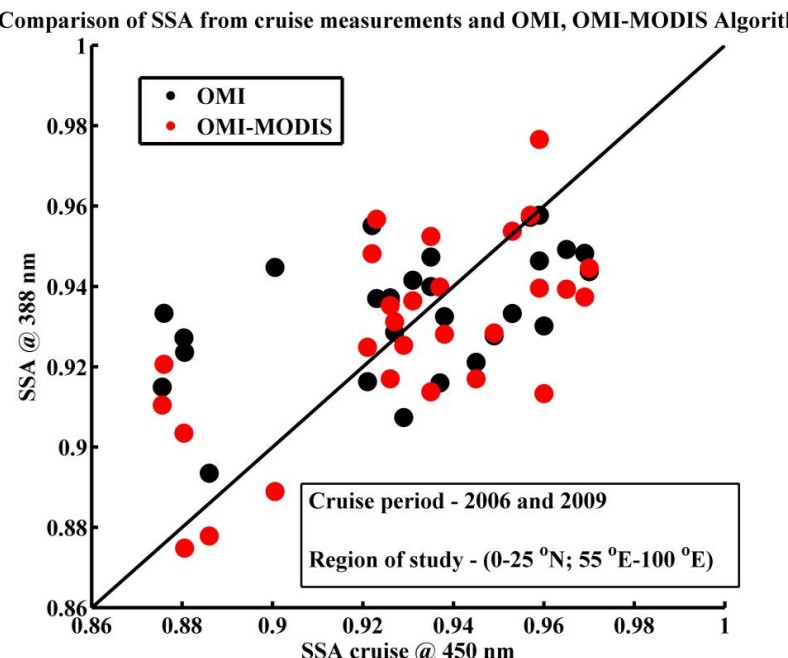





**Figure 10.** Comparison of SSA_OMI, SSA_OMI-MODIS with cruise measurements spatially averaged

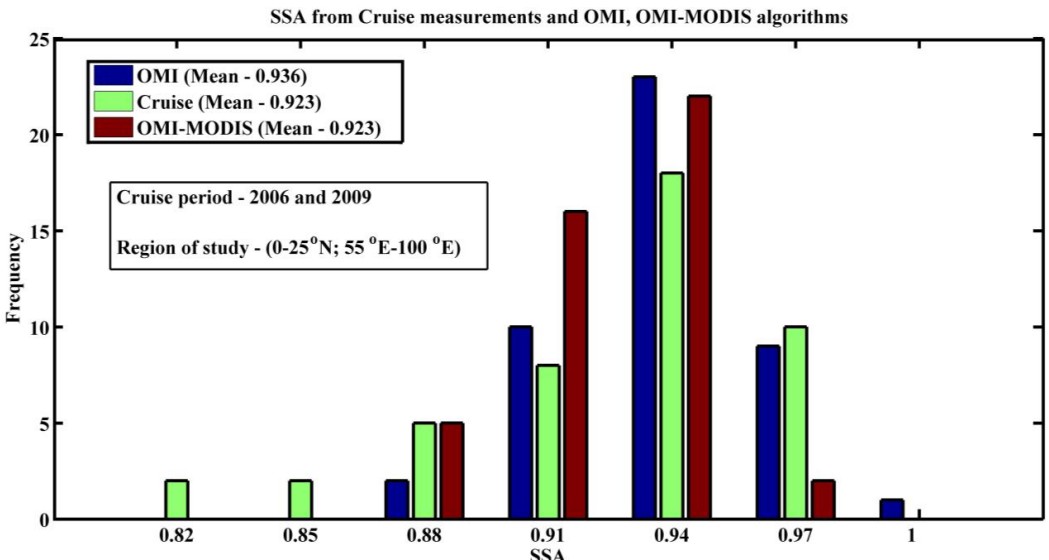


**Figure 11.** Distribution of SSA from OMI-MODIS, OMI and cruise measurements.

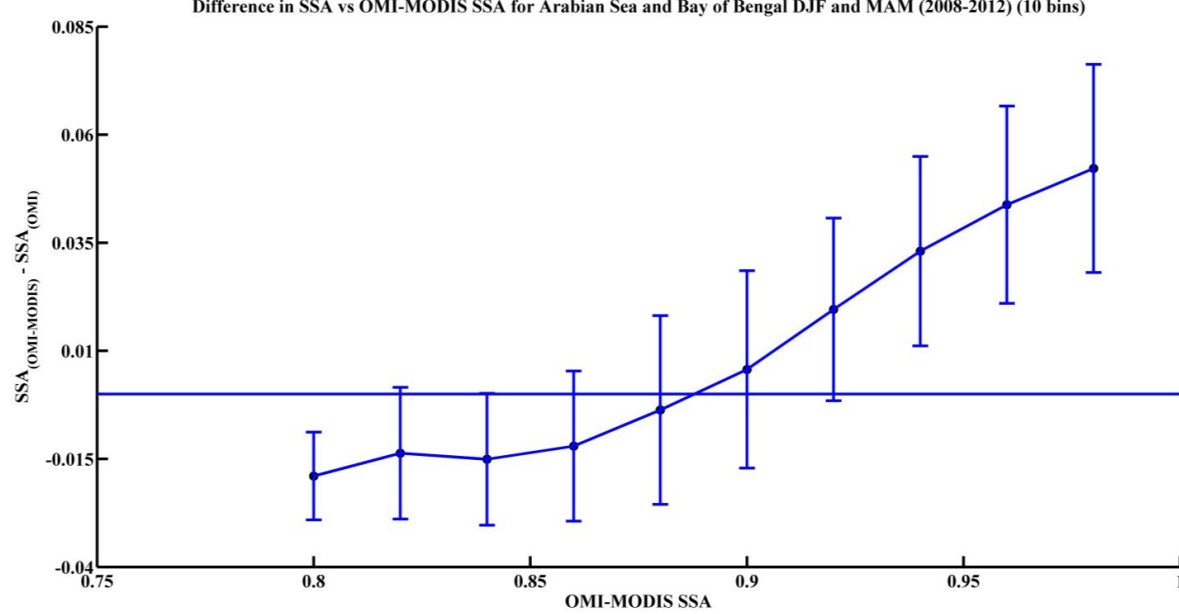


**Figure 12.** Difference in SSA from OMI-MODIS and OMI Vs SSA from OMI-MODIS. OMI



overestimates SSA when absorbing aerosols are detected by OMI-MODIS.





| References | Method | Technique | Limitation |
|---|---|---|---|
| Herman et al., 1975; King, 1979; Eck et al., 1998; Dubovik and King, 2000; Torres et al., 2005 | Ground-based observations | Inverse methods measurements of solar radiances and/or aerosol properties along with radiative transfer calculations | Measurements are spatially and temporally constrained |
| Dubovik et al., 2002 | Global network – Aerosols Robotic Network (AERONET) | Inverse technique using near-real time measured direct and diffuse radiation | Only land-based, low coverage over remote oceanic regions |



| | | | |
|---|---|---|---|
| Kaufman, 1987; Zhu et al., 2011; Wells et al., 2012 | Critical surface reflectance - where the net role of aerosol absorption and scattering becomes independent of aerosol optical thickness and is affected only by SSA | Over varying surface reflectance, the radiance difference between clear and hazy skies is measured using satellite images | Limited spatial variability of surface reflectance. Works only for few cases where there are large amount absorbing aerosols present |
| Kaufman et al., 2002b | Retrieve SSA in visible wavelengths | Sun-glint is used as a bright background to differentiate role of scattering from aerosol absorption | Only limited scenarios present and does not work on land when absorbing aerosols are present (Torres et al., 2005). |
| Diner et al., 1998; Remer et al., 2005 | Multi Angle Imaging Spectroradiometer (MISR) and Moderate Resolution Imaging Spectroradiometer (MODIS) | Retrieves AOD and SSA in the visible and infrared region of solar spectrum | Surface reflectance influences the retrievals |





| Herman et al., 1997; Torres et al., 1998 | Total Ozone Mapping Spectrometer (TOMS) | Aerosol index parameter is highly sensitive to the Rayleigh scattering thus acting as a bright background in the UV regime | Large pixel size prone to cloud contamination |
|---|---|---|---|
| Torres et al., 2002 | Ozone Monitoring Instrument (OMI) | Similar technique as TOMS. Pre-defined aerosol models used. | Sensitive to aerosol layer height and still prone to cloud contamination |


**Table 1.** Ground-based and Satellite-based indirect methods to retrieve SSA


| Seasons \ Regions | | 1 | | 2 | | 3 | | 4 | |
|---|---|---|---|---|---|---|---|---|---|
| | | 2009 | 2010 | 2009 | 2010 | 2009 | 2010 | 2009 | 2010 |
| DJF | 500m | 38% | 40% | 0% | 0% | **57%** | **45%** | 5% | 15% |
| | 1500m | **43%** | **45%** | 0% | 0% | 24% | 20% | 33% | 35% |
| | 2500m | **52%** | **50%** | 10% | 10% | 5% | 15% | 33% | 25% |
| MAM | 500m | 9% | 19% | 0% | 0% | **86%** | **62%** | 5% | 19% |
| | 1500m | **33%** | **38%** | 0% | 4% | **53%** | **29%** | 14% | 29% |
| | 2500m | **38%** | **24%** | 19% | 0% | 29% | 33% | 14% | 43% |





| | | | | | | | | |
|---|---|---|---|---|---|---|---|---|
| JJA | 500m | 5% | 5% | 0% | 0% | **90%** | **90%** | 5% | 5% |
| | 1500m | 9% | 5% | 0% | 0% | **67%** | **76%** | 24% | 19% |
| | 2500m | 0% | 5% | 0% | 0% | **76%** | **76%** | 24% | 19% |
| SON | 500m | 5% | 5% | 0% | 0% | **86%** | **71%** | 9% | 24% |
| | 1500m | 0% | 10% | 0% | 0% | **81%** | **71%** | 19% | 19% |
| | 2500m | 10% | 14% | 0% | 0% | **71%** | **57%** | 19% | 29% |

**Table 2.** Influence of various aerosol sources over Atlantic Ocean given as percentage of trajectories originating from each source respectively. The maximum influence is given in black bold. The different source regions are explained in text and Fig. 4.

| Seasons \ Regions | | 1 | 2 | 3 | 4 |
|---|---|---|---|---|---|
| DJF | 500m | **57%** | 0% | 38% | 5% |
| | 1500m | **62%** | 10% | 19% | 9% |
| | 2500m | **81%** | 14% | 0% | 5% |
| MAM | 500m | 19% | **43%** | 19% | 19% |
| | 1500m | 29% | **29%** | 23% | 19% |
| | 2500m | **57%** | 14% | 24% | 5% |
| JJA | 500m | 0% | 24% | 0% | **76%** |
| | 1500m | 19% | **67%** | 0% | 14% |
| | 2500m | **62%** | 33% | 5% | 0% |




| | | 1 | 2 | 3 | 4 |
|---|---|---|---|---|---|
| | 500m | 5% | 24% | **47%** | 24% |
| SON | 1500m | 14% | 19% | **48%** | 19% |
| | 2500m | **38%** | 10% | 19% | 33% |


**Table 3.** Influence of various aerosol sources over Arabian Sea given as percentage of
trajectories originating from each source respectively. The maximum influence is given in black
bold. The different source regions are explained in text and Fig. 5.

| Seasons＼Regions | | 1 | 2 | 3 | 4 |
|---|---|---|---|---|---|
| | 500m | **72%** | 0% | 14% | 14% |
| DJF | 1500m | **48%** | 14% | 10% | 28% |
| | 2500m | 29% | 33% | 0% | **38%** |
| | 500m | 19% | **48%** | 0% | 33% |
| MAM | 1500m | **57%** | 29% | 20% | 14% |
| | 2500m | **71%** | 24% | 0% | 5% |
| | 500m | 0% | **100%** | 0% | 0% |
| JJA | 1500m | 5% | **95%** | 0% | 0% |
| | 2500m | 14% | **81%** | 0% | 5% |
| | 500m | 5% | **52%** | 33% | 10% |
| SON | 1500m | 5% | **43%** | **43%** | 9% |
| | 2500m | 5% | **33%** | 29% | **33%** |




**Table 4.** Influence of various aerosol sources over Bay of Bengal given as percentage of trajectories originating from each source respectively. The maximum influence is given in black bold. The different source regions are explained in text and Fig. 6.


|  | **OMI** | **OMI-MODIS** |
|---|---|---|
| Mean SSA (Cruise – 0.923) | 0.936 | 0.923 |
| Std. Dev. (Cruise – 0.04) | 0.021 | 0.021 |
| p- value | 0.046 | 0.981 |
| Confidence Interval | [0.0002, 0.027] | [-0.013,0.013] |


**Table 5.** Comparison of SSA between both the satellite algorithms and cruise measurements
