# Peer review of "Multi-Satellite Retrieval of SSA using OMI-MODIS algorithm"

_Atmospheric Chemistry and Physics, 2018_

## Referee Comment (RC1) · Anonymous Referee #2 · 13 Aug 2018

Comments on acp-2018-564

Dear Editor,

Eswaran et al. paper submitted to AMT presents the new results on the retrieval of aerosol single-scattering albedo (SSA) over the global ocean derived from OMI-MODIS synergy algorithm originally developed by Satheesh et al. [2009]. The synergy algorithm takes advantage of better cloud screening of MODIS AOD retrievals combined with the better sensitivity of the near-UV spectrum to infer the aerosol SSA. The methodology of the retrieval essentially relies on that already published in Satheesh et al. [2009]. The author finds that OMI tends to overestimate SSA when the layer of absorbing aerosol resides closer to surface. The discrepancies in the SSA between the present retrieval dataset and that reported in the operational OMI/OMAERUV aerosol

product are attributed to the data discontinuity in the aerosol layer height climatology dataset used by the OMAERUV algorithm. The SSA retrieval dataset derived from the present algorithm has been claimed to be in better agreement when compared against the equivalent cruise-based measurements.

The content presented in the paper assumes important in estimating the radiative effects of aerosols for which accurate knowledge of aerosol SSA and layer height is a prime requirement along with the total aerosol loading.

A similar study carried out by Gasso and Torres [2016] presents the results on deriving SSA and ALH from OMI-MODIS synergy and discusses about the role of cloud contamination in OMI aerosol retrievals, which author misses to explain in greater details. A discussion highlighting important findings of Gasso and Torres [2016] and its (in)consistency with the new results presented in the paper is required. The paper requires some significant revisions at several places identified in the specific comments listed below and needs finishing on the language and presentation aspects, where I think both senior-level co-authors can step in and help to improve the manuscript further.

The author fails to mention which version of the OMI aerosol product they are using in the present research. I assume here that the latest OMAERUV version 1.8.9 has been employed here to derive the results. If not, the author needs to redo the entire analysis with the most recent dataset which incorporates a number of changes applied to the previous version of the algorithm.

I would be available to review the revised manuscript once authors provide their detail response to the following concerns with necessary changes included in the revision. Please share this letter with authors; it will be helpful in the improvisation of the submitted the manuscript.

Thanks,

Specific comments:

A similar study carried out by Gasso and Torres [2016] presents the results on deriving SSA and ALH from OMI-MODIS synergy and discusses the role of cloud contamination in OMI aerosol retrievals, which author misses to explain in greater details. A discussion highlighting important findings of Gasso and Torres [2016] and its (in)consistency with the new results presented in the paper is required.

Citation: Gassó, S. and Torres, O.: The role of cloud contamination, the aerosol layer height and aerosol model in the assessment of the OMI near-UV retrievals over the ocean, Atmos. Meas. Tech., 9, 3031-3052, https://doi.org/10.5194/amt-9-3031-2016, 2016.

Which version of the OMAERUV product does author use in the present study? I assume here that the latest OMAERUV version 1.8.9 has been employed here to derive the results. If not, the author needs to redo the entire analysis with the latest dataset which incorporates a number of changes applied to the previous version of the algorithm.

Introduction: A statement or two citing importance of SSA in estimating radiative forcing with appropriate reference would be needed in the introduction. Figure 5 can be used as a reference. Line 52-55: Author has completely forgotten to mention about the long-term record of aerosol absorption retrievals from AERONET! Line 55: Remove "images". Line 63-65: It is assumed here that the author is referring to the AOD retrievals from satellites. In this case, the statement "they still have a limited success over deserts" is untrue; AOD retrievals over bright surfaces from the near-UV, Deep Blue, and MAIAC algorithms have achieved a great success in retrieving accurate AODs. Line 67-69: What do the "large surface reflectance contrasts" means? Line 72: The sub-pixel cloud contamination is a result of the larger footprint of size 13 x 24 km-square. Line 76: The time difference between the observations from OMI and MODIS is about 7-8 minutes post-2008 period. A brief discussion about the OMI-MODIS com-

bined retrieval approach is needed here.

Section 2.1 First paragraph: In addition to the higher sensitivity to aerosol loading and its absorption properties, the 354 and 388 nm wavelengths have negligible interference from trace gases. "sulphate-based" aerosol type was a gross terminology used for the boundary layer aerosols; it should be changed to "background and urban-industrial" aerosol type. Line 111: Rephrase the sentence as "the retrievals are performed reported for the five discrete aerosol layer heights, i.e., surface (exponential profile), 1.5, 3.0, 6.0, and 10.0 km with latter four following a Gaussian distribution. The final set of AOD/SSA/AAOD retrievals is reported at the mean ALH provided by the 30-month long averaged climatology developed using OMI-CALIOP combined observations." Line 120: "have helped distinguish carbonaceous aerosols from dust particles" Line 126: "An effective aerosol layer height was calculated using the CALIOP 1064 nm attenuated backscatter weighted by corresponding altitudes" Line 128: "..in the OMAERUV retrievals which then validated against the AERONET observations"

3. Algorithm MODIS aerosol product reports retrievals at 10 x 10 km spatial resolution at nadir. Line 160: "..prone to sub-pixel cloud contamination which may result in overestimation in AOD and SSA" Line 163: The high accuracy of size-resolved aerosol retrievals with MODIS is because the over-ocean algorithm employs all seven channels (0.47-2.13 micron) in the inversion enabling better characterization of fine and coarse particles. Line 169: "..constraints the retrievals of AOD and SSA" Line 179: Does the algorithm use Angstrom Exponent retrieved by MODIS over the ocean?

Section 5. Results Line 214-215: not a phenomenon, but the instrumental issue. While a brief discussion on row anomaly is provided in Jethva et al. [2014], Torres et al. [2018] discuss it in great details and its effect on the scan dependency on the OMAERUV retrievals. Citation: Torres, O., Bhartia, P. K., Jethva, H., and Ahn, C.: Impact of the ozone monitoring instrument row anomaly on the long-term record of aerosol products, Atmos. Meas. Tech., 11, 2701-2715, https://doi.org/10.5194/amt-11-2701-2018, 2018.

Section 5.1 The differences could also be attributed to the shape of the dust particle. In the latest OMAERUV (V1.8.9) algorithm, dust is assumed to be of spheroidal shape with axis ratio distribution adopted from Dubovik et al [2006] study. Please refer to Torres et al. [2018] AMT paper; the citation is provided earlier in this report.

Section 6. Summary and conclusion In addition to the sub-pixel cloud contamination and ALH, an uncertainty in prescribing surface albedo is another source of error in the retrieval of SSA from space. 3. Is OMI unable to retrieve absorbing aerosols for low ALH or does retrievals but under/over-estimate SSA? 5. Provide the statistics of the cruise vs. satellite comparison.

Figures: Figure 1. Since the dynamical range of SSA variations in these maps is confined to 0.85-1.0, a narrower color scale covering this range would be desirable. Figure 2. For the most part, SSA retrieved from OMI-MODIS synergy is larger than that retrieved from OMI only. Figure 4. From where does the author get ALH difference of 8-10 km? Figure 8. Add the measures of agreement, i.e., N, RMSD, correlation

A plot demonstrating the effect of a change in AOD on the change in SSA either through radiative transfer calculations or from satellite data is needed here. The author may choose a representative region here, say the tropical Atlantic Ocean with dust transport from Sahara for such analysis.

---

## Referee Comment (RC2) · Anonymous Referee #1 · 21 Sep 2018

First of all I need to apologize to the authors and to the editor for not understanding that I was supposed to review this manuscript and for the late return of the review.

I will summarize my overall impression of the manuscript.

The authors apply a method developed by Satheesh et al. (2009) that uses MODIS retrievals to constrain the AOD in an OMI retrieval of aerosol layer height (ALH) and single scattering albedo (SSA). Currently the operational OMI retrieval uses climatology to constrain ALH and thus retrieves AOD and SSA. The point is that OMI has three variables and only two pieces of information. Something has to be constrained or assumed. The authors apply the OMI-MODIS retrieval over the global oceans and compare with the operational OMI product. The global study points them to two regions of particular interest: the tropical Atlantic and the seas surrounding the Indian

subcontinent. They want to know more about the aerosol in these regions so they pick a point in each region and do a lot of trajectory analysis to find out from where the air mass originates at three different altitudes. I am still unclear on how the trajectory analysis ties into the comparison of the two different retrieval methods. After the trajectory analysis they go back to comparing the results of the two methods. Here they find that when OMI-alone underestimates SSA, it also underestimates ALH. Because they believe the OMI-MODIS retrieval is more "truth" than OMI alone, they conclude that OMI-alone underestimates SSA when the aerosol is at low altitude. Note, we haven't seen any validation yet. Then they run some radiative transfer and rediscover the interplay between aerosol absorption, AOD and ALH, all because of Rayleigh scattering. Then we compare SSA retrievals with in situ measurements made during a cruise in the Bay of Bengal, where we find that both retrieval methods match the cruise data equally well within their respective uncertainties, but that the OMI-MODIS retrieval exhibits less over all mean bias.

For the most part the writing is good enough for me to understand the authors' intent. The exceptions are noted in the line by line analysis below. However, the typical grammar errors expected of new-to-English writers do permeate the manuscript. I will attach an annotated pdf that corrects some, but not all of the English problems.

In some ways this manuscript represents a lot of very good work in search of a paper. What is this paper? 1. A proposal to use the MODIS-OMI retrieval instead of the operational OMI retrieval? 2. A picture of the global (or regional) SSA, as retrieved by this new method, assuming that it is already established as a better method? 3. An attempt to better understand the regional aerosol system in the north tropical Atlantic, Arabian Sea and Bay of Bengal? Right now it takes steps in each of these 3 directions without really succeeding at any of them because the reader is pulled in multiple directions. The paper needs a rethinking and a rewrite, but the work itself is worthy of publication (with the possible exception of the re-discovery of the importance of Rayleigh scattering). I would recommend either Major Revision or Reject with the Encouragement to

Resubmit.

Initially I worried that this work may duplicate the Gassó and Torres (2016) paper, but it does not. It is very different, and the authors do an excellent job of putting their work into context with this previous work. Be sure to put the accent over the 'o' in Gassó when citing.

Here are my line-by-line comments. The most important are indicated by ******.

Line 30. "Forcing" or "effect". Some people use forcing only when the aerosols are anthropogenic. Clarification in definition here. IPCC reference to 'forcing' is only anthropogenic and that is where that statement is leading later on in the paragraph.

Line 40-41. 'the fraction of the total extinction of radiation attributed to scattering'

Line 44-45. "However, SSA values lack high certainty (Bond and Bergstrom, 2006; Bond et al., 2013)" What has high uncertainty? Measurements of SSA? Attributing SSA to different aerosol types? Understanding the overall SSA of aerosols globally or regionally? "SSA values" is too ambiguous.

Line 44-51. Lots of ambiguities here between measurements, retrievals and physical properties.

Lines 52-72. *****While Table 1 is very good and a major contribution of the paper by itself. This paragraph needs clarification between "direct" and "indirect" measures of SSA. Again, what is a measurement? What is a retrieval? What are the pluses and minuses of each? I see that in the next paragraph some of this explanation is attempted, but the organization of the whole delivery is confusing.*****

Figure 1 caption. State the wavelength.

Line 214-215. Some places on the globe will not have a lot of retrievals because AOD is usually low and there is an AI criteria as to when to retrieve. There might only be one retrieval in that grid box in 5 years. Do the plots in Fig. 1 show points like these?

Is that a fair representation of the climatology?

Lines 217-219. Are these statements based on Figure 1 or some previous work or understanding? Because they don't match what I see in Figure 1. Even if you ignore the tropical Atlantic because of dust, I see a lot of SSA in the 0.9 to 0.95 range in the open oceans and near land, I don't see anything that gets lower than 0.85. Where is the 0.75?

Line 220-221. From my own studies based on AOD_550, not AI, the threshold is AOD_550 = 0.30. Greater than that and I don't see the ocean anymore.

Lines 238-240. What happens when one method has a value and the other method does not? This should be stated in the text, and possibly the caption to Figure 2. This in itself is of a lot of interest to people. *****Why is OMI retrieving so much more than OMI-MODIS?**** You mention OMI is cloud contaminated and MODIS is not. Is this difference in number of retrievals due to cloud masking? Can you prove that? *****Because the cloud masking issue is never addressed anywhere in the paper.*****

Figure 2 caption. We need the wavelength of the SSA, and in the caption, it should tell us that it is OMI-MODIS minus OMI. It should also tell us what happens when one product has values and the other does not.

Line 245-247. What are natural aerosols here and what are anthropogenic? Dust and smoke? Please clarify. Also, at least at this point it's not easy for me to see how differences in the method results are linked to actual aerosol properties. The speculation here seems premature. Most importantly, rather than dwelling on the differences in aerosol types/properties the text should mention problems with the height assumption. That would be my first guess as to what I'm seeing here, not aerosol types. Also the differences are relatively small, within what I would expect to be the uncertainty in any satellite retrieval of SSA.

Figure 4. That isn't southern Africa. Maybe call it Central Africa? Same for Figure 5.

Table 2 and 3. Don't use numbers for regions. Use their names. Also, this is not for the "Atlantic Ocean", but for one specific point in the Atlantic Ocean. Likewise for the Arabian Sea.

Lines 286-291. "Harriss et al. (1984), found that there is advection of anthropogenic pollutants from North America to the North Atlantic Ocean." I don't have time to look up that reference, but does it include that one point at 15N 45 W? Also 1984 is a long, long time ago. Aerosols in North America have changed significantly since then and your study period is 2009-2010. Also there is no reference on the NOAA-11 study. 1988-2004. That's a bit better in terms of matching this paper's study period, but not much.

Lines 299-300. I can visualize, maybe, a large scale circulation that is creating westerlies aloft during winter and spring at that point. It would have to be the winter time baroclinic systems dipping far south. The question though is that at least in winter there would be no aerosol associated with that flow. Spring time you may be getting biomass burning from Mexico. ****It would be useful to better describe the meteorology affecting the situation.*****

Lines 302-331. The meteorological description here is much better than that over the Atlantic. Here, a single point in the middle of the Arabian Sea is better representative of the entire region than a single point in the north tropical Atlantic trying to represent the entire "Atlantic Ocean". But also the authors just convey a much clearer understanding of the meteorological and aerosol forces influencing that point in the Arabian sea than they do in the north tropical Atlantic.

Lines 332 -336. These sentences are so convoluted I don't understand the point the authors are trying to make.

Lines 355-357. Here the terms natural and anthropogenic are being used without really defining them.

****Trajectory analysis overall. I don't see how all this work connects to the rest of the paper.****

Lines 362-365. From the histograms I don't see much differences between the Atlantic and the Arabian Sea in terms of how well the results from the two methods match. What is considered "reasonably good agreement"?

Line 363. Can you remind the reader which season is the dust season? From the histograms, it looks like MAM is the least biased season, and that is not the dust season, right?

Line 365, but I do agree that height should be the important factor, not aerosol type.

Lines 367-369. I'm not sure what is meant here. In this work the ALH is calculated for OMI using the best estimate of SSA retrieved from OMI. This is the operational OMI-only retrieval we are talking about, not the OMI-MODIS retrieval, right? How is the best estimate SSA determined? This retrieval returns 5 ordered pairs of (SSA, ALH) and the retrieval fixes ALH and returns SSA. Fine. Now, in this work, the authors are going to fix SSA and return ALH. Ok. But. . . how do they decide on an SSA? The caption for Figure 7 explains it, but the text should match.

Figure 7. Very good and informative caption. They should all be this good.

Lines 372-373. This assumes that the OMI-MODIS retrieval is correction, which has not been proven. The wording is also awkward for me. What I would say is this: The most important observation from this analysis is that the operational OMI-only retrieval of SSA overestimates SSA when it also overestimates ALH, and vice-versa.

Lines 374-379. Does it matter whether or not the operational OMI uses CALIPSO climatology or the prior assumptions? Did you study this? I don't think so. The algorithm isn't using real-time collocated CALIPSO. It is using CALIPSO climatology. There could still be issues. Anyway, because you didn't actually study the difference between CALIPSO climatology and prior climatology, these details here are just distracting.

Lines 379-381. This sentence is very good and valuable.

Lines 382-390. I don't know understand the point the authors are trying to make here. The paragraph wanders.

Line 396 is unfinished

Lines 391-404, and Figures 8 and 9. These are not earth-shattering results. We all know this. I don't have time to look back into the old Deep Blue papers, but this is the basis for that algorithm. *****I'm not opposed to including this analysis in the paper, but it has to be put into context with previous work. **** Also, I might combine Figures 8 and 9 into a single 2-panel figure.

Figure 10 caption needs a lot more detail. What does each point represent in terms of spatial/temporal averaging? What is the correlation? Is there any correlation?

Line 428. I think it is an accident that the MODIS-OMI mean matches the cruise exactly. The statistics tell the same story that I see with my eye. . . The two retrievals match the cruise about the same, to within their expected uncertainties.

Section 5.4 as a whole. It's dangerous to expect the total column ambient retrievals to match whatever was making in situ measurements at the ocean surface. Different everything. Some of these caveats need to be expressed in this section. ****Also and this is critical. . . we need to know what instrument was used on the cruise and exactly what it measured. What wavelength? What method? Did it dry aerosols or not? The name of the ship. Other things. Details here are essential.****

Lines 457-458, or chances in ALH as the SAL cools and descends, right? I saw that gradient and I thought ALH right away, not changing aerosol properties.

Line 459. "OMI overestimates SSA at lower ALH and underestimates at higher values of ALH." Sure, if the OMI-MODIS is true.

Lines 459-463. ****Again, I don't think you can say anything about the differences

between CALIPSO climatology versus prior climatology. This should not be here in the major conclusions. What I might say here is, "Despite the operational algorithm moving to CALIPSO climatology, we continue to find systematic differences in the algorithm's SSA-ALH retrieval, when compared with the more robust OMI-MODIS retrieval. This may be due to situations when CALIPSO climatology is missing and the algorithm reverts to prior assumptions, or more likely, it may be due to lingering uncertainties in ALH even when using the improved climatology."****

Line 464-466. ****Again, we all already know this. It is strange to find it in the major conclusions.****

Lines 467-470. ****I think you are writing the way you wished it turned out. What you actually found that there was no significant difference between the OMI and OMI-MODIS retrieval in matching the cruise data, although the overall mean OMI-MODIS SSA for the area and period showed virtually no bias against the cruise data, while the OMI-only retrieval mean was biased 0.013 too high.*****

Lines 471-472. I'm not sure about this point at all.

Line 474-475. ****What makes you say that the OMI-MODIS is able to detect absorbing aerosols much better than OMI? Detect is not the same as retrieving SSA. Keep that in mind. Note also in the global maps OMI has much better coverage than OMI-MODIS. Why? You never discussed that and it's important. Is OMI reporting cloud contaminated results? Or is OMI much better at detection? *****

Please also note the supplement to this comment:
https://www.atmos-chem-phys-discuss.net/acp-2018-564/acp-2018-564-RC2-supplement.pdf

**Supplement:**

[revised manuscript text omitted]

---

## Author Comment (AC1) · 6 Dec 2018

**Author response to referee comments for the manuscript titled "Multi-Satellite Retrieval of SSA using OMI-MODIS algorithm"**

We thank both the referees for their valuable comments and suggestions in improving this manuscript. Please note that author comments (AC) are in red font color. The entire analysis has been redone using the latest version of OMI (v1.8.9). The modified manuscript along with the author marked changes are provided after the authors' response.

**Anonymous Referee #1**

Here are my line-by-line comments. The most important is indicated by ******.

RC: Line 30. "Forcing" or "effect". Some people use forcing only when the aerosols are anthropogenic. Clarification in the definition is needed here. IPCC reference to 'forcing' is only anthropogenic and that is where that statement is leading later on in the paragraph.

*AC: While the aerosol forcing definition according to IPCC is due to only anthropogenic aerosols, for the present study it is defined as the combined effect of both natural and anthropogenic aerosols.*

RC: Line 40-41. 'the fraction of the total extinction of radiation attributed to scattering'

*AC: The statement has been modified accordingly.*

RC: Line 44-45. "However, SSA values lack high certainty (Bond and Bergstrom, 2006; Bond et al., 2013)" What has high uncertainty? Measurements of SSA? Attributing SSA to different aerosol types? Understanding the overall SSA of aerosols globally or regionally? "SSA values" is too ambiguous.

*AC: The sentence has been corrected as measurements of SSA.*

RC: Line 44-51. Lots of ambiguities here between measurements, retrievals and physical properties.

*AC: We thank the reviewer for pointing out the mistake. The phrase used is SSA retrievals.*

RC: Lines 52-72. *****While Table 1 is very good and a major contribution of the paper by itself. This paragraph needs clarification between "direct" and "indirect" measures of SSA. Again, what is a measurement? What is a retrieval? What are the pluses and minuses of each? I see that in the next paragraph some of this explanation is attempted, but the organization of the whole delivery is confusing. *****

*AC: We thank the reviewer for pointing out the mistake. The words 'direct' and 'indirect' are removed. The statement has been modified as follows "Studies on the various measurements of aerosol absorption using instruments and their uncertainty evaluation have been performed previously (Horvath, 1993, Heintzenberg et al., 1997; Moosmuller et al., 2009).*

*Along with ground-based retrievals of SSA, there have been other methods to retrieve the parameter using satellites (Table 1). The different methods of retrieval of SSA, both ground-based and using satellites, are provided in Table 1. Unlike aerosol absorption coefficient, SSA is not measured directly by an instrument. Instead, it is retrieved using lookup tables or estimated using other parameters which are measured or calculated using models."*

RC: Figure 1 caption. State the wavelength.

*AC: The wavelength has been added.*

RC: Line 214-215. Some places on the globe will not have a lot of retrievals because AOD is usually low and there is an AI criterion as to when to retrieve. There might only be one retrieval in that grid box in 5 years. Do the plots in Fig. 1 show points like these? Is that a fair representation of the climatology?

*AC: Yes, there are grid boxes which have only one retrieval in 5 years. Such boxes are not preferred as a climatological representation. If these boxes were removed, the number of points reduced drastically. However, Fig.1 is not considered as a climatology plot since it involves only five years of data. It is plotted to understand the spatial coverage and variation of both the algorithms for the different seasons over the five years. For better representation of climatology data over more years are required.*

RC: Lines 217-219. Are these statements based on Figure 1 or some previous work or understanding? Because they don't match what I see in Figure 1. Even if you ignore the tropical Atlantic because of dust, I see a lot of SSA in the 0.9 to 0.95 range in the open oceans and near land, I don't see anything that gets lower than 0.85. Where is the 0.75?

*AC: A narrower colour scale from 0.85 to 1 has been used to represent global SSA maps.*

RC: Line 220-221. From my own studies based on AOD_550, not AI, the threshold is AOD_550 = 0.30. Greater than that and I don't see the ocean anymore.

*AC: We agree with the referee that over oceans AOD does not exceed 0.3 but we are talking about AI, not AOD*

RC: Lines 238-240. What happens when one method has a value and the other method does not? This should be stated in the text, and possibly the caption to Figure 2. This in itself is of a lot of interest to people. *****Why is OMI retrieving so much more than OMI-MODIS?**** You mention OMI is cloud contaminated and MODIS is not. Is this difference in the number of retrievals due to cloud masking? Can you prove that?*****Because the cloud masking issue is never addressed anywhere in the paper.*****

*AC: In the OMI-MODIS algorithm, the aerosol layer height was retrieved through linear interpolation of $AOD_{OMI}$ at five different heights and $AOD_{388}$ as a reference. Linear interpolation was not performed for OMI retrievals which had a missing value at any particular height or if the OMI retrieval was the same at all heights, resulting in the final OMI-MODIS value to be invalid. Similarly, if the MODIS AOD was found to be missing or invalid, the corresponding OMI-MODIS retrieval was also considered invalid. The removal of such invalid retrievals resulted in a reduction in the total number of valid points in OMI-MODIS algorithm when compared to OMI algorithm (Fig. 1b).*

*Gassó and Torres (2016) for a particular day over the North Central Atlantic compared the AOD values retrieved by OMI and MODIS. They compared the difference with the aerosol cloud mask retrieved by MODIS. It was found that while most of the retrievals of OMI screened the cloudy pixels, some of the best quality (flag=0) pixels were found to be cloud contaminated. They attributed this to the coarser pixel size of OMI compared to the smaller pixel size of MODIS cloud product. Using the MODIS cloud fraction to screen out OMI cloudy pixels improved the agreement between AOD values but resulted in a reduction in the number of OMI retrievals despite good agreement between the AOD values. In some cases, MODIS showed large cloud fraction values when the aerosol index was high implying the presence of aerosols above clouds. Hence Gassó and Torres concluded that only MODIS cloud fraction could not be used to screen out OMI pixels. Such an analysis is needed on a larger spatiotemporal scale which is beyond the scope of this manuscript and will be considered in a separate work.*

RC: Figure 2 caption. We need the wavelength of the SSA, and in the caption, it should tell us that it is OMI-MODIS minus OMI. It should also tell us what happens when one product has values and the other does not.

*AC: The wavelength has been added. When the OMI-MODIS SSA value was found invalid, the difference was also considered to be invalid.*

RC: Line 245-247. What are natural aerosols here and what are anthropogenic? Dust and smoke? Please clarify. Also, at least at this point it's not easy for me to see how differences in the method results are linked to actual aerosol properties. The speculation here seems premature. Most importantly, rather than dwelling on the differences in aerosol types/properties the text should mention problems with the height assumption. That would be my first guess as to what I'm seeing here, not aerosol types. Also the differences are relatively small, within what I would expect to be the uncertainty in any satellite retrieval of SSA.

*AC: As suggested by the reviewer the statement has been modified as follows "This was attributed to the change in aerosol layer height and (or) aerosol physical and optical properties." The different aerosol sources during different seasons were studied through trajectory analysis to understand the role of aerosol type in the change of SSA between algorithms. Following this discussion, the role of aerosol layer height in SSA retrieval was examined. The study by Torres et al. (2018) on the latest version of OMI aerosol product shows that the shape of dust aerosols assumed also affected the SSA retrievals implying the importance of knowing the aerosol type along with its physical properties for the retrieval of SSA.*

RC: Figure 4. That isn't southern Africa. Maybe call it Central Africa? Same for Figure 5.

*AC: The trajectory analysis for the Atlantic Ocean has been removed since it was found that the difference of SSA retrievals between the two algorithms over this region was within ±0.03.*

RC: Table 2 and 3. Don't use numbers for regions. Use their names. Also, this is not for the "Atlantic Ocean", but for one specific point in the Atlantic Ocean. Likewise, for the Arabian Sea.

*AC: The numbers are used only on the plot for ease in reading. The regions representing the numbers are mentioned in the caption as well as in the text. As suggested by the reviewer, the trajectory analysis was indicated for one specific point.*

RC: Lines 286-291. "Harriss et al. (1984), found that there is advection of anthropogenic pollutants from North America to the North Atlantic Ocean." I don't have time to look up that reference, but does it include that one point at 15N 45 W? Also 1984 is a long, long time ago. Aerosols in North America have changed significantly since then and your study period is 2009-2010. Also there is no reference on the NOAA-11 study. 1988-2004. That's a bit better in terms of matching this paper's study period, but not much.

*AC: Since the difference in SSA between the algorithms was within the ±0.03 range over the Atlantic Ocean, the trajectory analysis was done only for a point over the Arabian Sea and the Bay of Bengal to understand the aerosol sources that affect the region during each season and the variation of difference in SSA for each season.*

RC: Lines 299-300. I can visualize, maybe, a large-scale circulation that is creating westerlies aloft during winter and spring at that point. It would have to be the wintertime baroclinic systems dipping far south. The question though is that at least in winter there would be no aerosol associated with that flow. Springtime you may be getting biomass burning from Mexico. ****It would be useful to better describe the meteorology affecting the situation. *****

*AC: The trajectory analysis for the Atlantic Ocean has been removed.*

RC: Lines 302-331. The meteorological description here is much better than that over the Atlantic. Here, a single point in the middle of the Arabian Sea is better representative of the entire region than a single point in the north tropical Atlantic trying to represent the entire "Atlantic Ocean". But also the authors just convey a much clearer understanding of the meteorological and aerosol forces influencing that point in the Arabian sea than they do in the north tropical Atlantic.

*AC: We thank the reviewer for the comment.*

RC: Lines 332 -336. These sentences are so convoluted I don't understand the point the authors are trying to make.

*AC: The statement has been modified as follows "While the Arabian Sea is dominated by dust and oceanic aerosols, studies have shown that the Bay of Bengal is influenced by various air masses associated with Asian monsoon system including those of anthropogenic origin."*

RC: Lines 355-357. Here the terms natural and anthropogenic are being used without really defining them. ****Trajectory analysis overall. I don't see how all this work connects to the rest of the paper. ****

*AC: It has been previously mentioned that "The IGP with its heavy population and a large number of industries acts as a source for anthropogenic aerosols which are transported to*

*the Bay of Bengal during winter". These along with the biomass aerosols are considered anthropogenic, and the sea-salt aerosols from nearby ocean and the dust from Arabian Peninsula and Indian subcontinent are considered natural aerosols. The trajectory analysis was performed to understand the seasonal variation of SSA of both the algorithms due to the difference in aerosol sources between seasons.*

RC: Lines 362-365. From the histograms I don't see much differences between the Atlantic and the Arabian Sea in terms of how well the results from the two methods match. What is considered "reasonably good agreement"?

*AC: The results mentioned are figures from Satheesh et al. (2009). In their study, they have compared MODIS extrapolated UV AOD with OMI AOD over the Atlantic Ocean and the Arabian Sea. They found that over the Atlantic on an average the AOD agreed within ±0.1. Over the Arabian Sea, they found agreement between both the AOD retrievals during the months when a large amount of dust aerosols is present (April-July).*

RC: Line 363. Can you remind the reader which season is the dust season? From the histograms, it looks like MAM is the least biased season, and that is not the dust season, right?

*AC: The dust season over the Arabian Sea is March-April-May. According to Satheesh et al. (2009), MODIS UV AOD and OMI AOD agreed well during this season when there is the large loading of dust.*

RC: Line 365, but I do agree that height should be the important factor, not aerosol type.

*AC: We agree with the reviewer that aerosol height is the main factor affecting the retrievals.*

RC: Lines 367-369. I'm not sure what is meant here. In this work the ALH is calculated for OMI using the best estimate of SSA retrieved from OMI. This is the operational OMI only retrieval we are talking about, not the OMI-MODIS retrieval, right? How is the best estimate SSA determined? This retrieval returns 5 ordered pairs of (SSA, ALH) and the retrieval fixes ALH and returns SSA. Fine. Now, in this work, the authors are going to fix SSA and return ALH. Ok. But: : : how do they decide on an SSA? The caption for Figure 7 explains it, but the text should match.

*AC: The OMI retrieval has been explained in section 2.1. Along with the aerosol products retrieved at different heights, the final set of SSA retrievals in the OMAERUV product are*

*reported at the mean ALH provided by the 30-month long averaged climatology developed using OMI-CALIOP combined observations (Torres et al., 2013). This mean climatological ALH is taken as the OMI algorithm ALH.*

RC: Figure 7. Very good and informative caption. They should all be this good.

*AC: We thank the reviewer for the comment.*

RC: Lines 372-373. This assumes that the OMI-MODIS retrieval is correction, which has not been proven. The wording is also awkward for me. What I would say is this: The most important observation from this analysis is that the operational OMI-only retrieval of SSA overestimates SSA when it also overestimates ALH, and vice-versa.

*AC: The sentence has been modified accordingly.*

RC: Lines 374-379. Does it matter whether or not the operational OMI uses CALIPSO climatology or the prior assumptions? Did you study this? I don't think so. The algorithm isn't using real-time collocated CALIPSO. It is using CALIPSO climatology. There could still be issues. Anyway, because you didn't actually study the difference between CALIPSO climatology and prior climatology, these details here are just distracting.

*AC: The details regarding the aerosol layer height assumption have been removed.*

RC: Lines 379-381. This sentence is very good and valuable.

*AC: We thank the reviewer for the comment.*

RC: Lines 382-390. I don't know understand the point the authors are trying to make here. The paragraph wanders.

*AC: The paragraph has been modified.*

RC: Line 396 is unfinished

*AC: The sentence has been modified.*

RC: Lines 391-404, and Figures 8 and 9. These are not earth-shattering results. We all know this. I don't have time to look back into the old Deep Blue papers, but this is the basis for that algorithm. *****I'm not opposed to including this analysis in the paper, but it has to be put into context with previous work. **** Also, I might combine Figures 8 and 9 into a single 2-panel figure.

*AC: The paragraph has been modified, and the figures have been combined. The modification is as follows "The basis of many aerosol retrievals by satellites in the UV spectrum is the sensitivity of aerosol absorption to Rayleigh scattering which acts as a bright background and contributes to the TOA radiance (Torres et al., 1998; 2002). Change in ALH can affect the TOA radiance since the aerosol layer will interact with the Rayleigh scattering due to molecules present in the atmosphere. However, this effect is smaller compared to the effect due to the change in AOD and SSA (Kim et al., 2018). Kim et al. (2018) also showed how the misclassification of aerosol type and size could affect ALH retrieval. OMI SSA retrievals which are based on LUT depend on the ALH assumed along with aerosol type. The SBDART simulations in the current work show how for a particular TOA flux, SSA varies with ALH when the other aerosol properties are kept constant."*

RC: Figure 10 caption needs a lot more detail. What does each point represent in terms of spatial/temporal averaging? What is the correlation? Is there any correlation?

*AC: The caption has been changed with more explanation*

RC: Line 428. I think it is an accident that the MODIS-OMI mean matches the cruise exactly. The statistics tell the same story that I see with my eye: : : The two retrievals match the cruise about the same, to within their expected uncertainties.

*AC: After using the new version (1.8.9) of OMI, section 5.4 has been rewritten.*

RC: Section 5.4 as a whole. It's dangerous to expect the total column ambient retrievals to match whatever was making in situ measurements at the ocean surface. Different everything. Some of these caveats need to be expressed in this section. ****Also and this is critical: : : we need to know what instrument was used on the cruise and exactly what it measured. What wavelength? What method? Did it dry aerosols or not? The name of the ship. Other things. Details here are essential.****

*AC: We thank the reviewer for his suggestion. The following has been added*

*"During both the cruises, the aerosol sampling was done onboard the Oceanic Research Vessel Sagar Kanya. While the 2006 cruise covered both the Arabian Sea and the Bay of Bengal, the winter cruise of 2009 covered the Bay of Bengal. The cruise tracks are provided in detail in Moorthy et al., 2008 and 2010, respectively. The SSA values at different wavelengths were estimated from spectral values of the absorption coefficient and scattering coefficient, measured using the instruments Aethalometer (Magee Scientific AE-31, USA) and*

*an integrating nephelometer (TSI 3563, USA) respectively. More details about the instrument and measuring techniques including the uncertainties are provided in Nair et al. (2008). However, both the cruise did not estimate SSA values in the UV spectrum. The closest wavelength at which SSA was calculated is 450nm which has been used to compare with the satellite retrievals of SSA (388nm). Ground-based SSA estimates based on in-situ measurements are seldom consistent with columnar retrievals from satellites especially when elevated aerosols are present. This uncertainty along with the uncertainty in the assumption of SSA being uniform between 388nm and 450nm implies that the current comparison of study cannot be used as a validation study. Instead, it is used to understand the consistency of SSA retrievals from satellites with ground-based observations."*

RC: Lines 457-458, or chances in ALH as the SAL cools and descends, right? I saw that gradient and I thought ALH right away, not changing aerosol properties.

*AC: The conclusion point has been modified as follows "The difference in SSA retrievals of both the algorithms (ΔSSA) was found to be within ±0.03 over ATL >80% of the time during all the seasons. Over the Arabian Sea, as seen in Satheesh et al. (2009), ΔSSA was within the ±0.03 range during MAM when the region was influenced by dust. The discrepancies during other season were due to the wrong assumption of aerosol layer height by OMI."*

RC: Line 459. "OMI overestimates SSA at lower ALH and underestimates at higher values of ALH." Sure, if the OMI-MODIS is true.

*AC: The statement has been modified as "seen that OMI overestimated SSA when it overestimated ALH and vice versa which can be attributed to the wrong assumption of aerosol height."*

RC: Lines 459-463. ****Again, I don't think you can say anything about the differences between CALIPSO climatology versus prior climatology. This should not be here in the major conclusions. What I might say here is, "Despite the operational algorithm moving to CALIPSO climatology, we continue to find systematic differences in the algorithm's SSA-ALH retrieval, when compared with the more robust OMI-MODIS retrieval. This may be due to situations when CALIPSO climatology is missing and the algorithm reverts to prior assumptions, or more likely, it may be due to lingering uncertainties in ALH even when using the improved climatology." ****

*AC: We thank for the reviewer for the suggestion. The point has been modified as follows "During winter, when the aerosols were present closer to the surface, OMI-MODIS was more consistent compared to OMI which may be due to scenarios where the CALIPSO climatology is absent and OMI uses its previous aerosol model assumptions. The difference could also be due to the uncertainties in ALH value even after the improvement in the OMI algorithm with the addition of CALIPSO climatology."*

RC: Line 464-466. ****Again, we all already know this. It is strange to find it in the major conclusions. ****

*AC: The conclusion point has been removed.*

RC: Lines 467-470. ****I think you are writing the way you wished it turned out. What you actually found that there was no significant difference between the OMI and OMI-MODIS retrieval in matching the cruise data, although the overall mean OMI-MODIS SSA for the area and period showed virtually no bias against the cruise data, while the OMI-only retrieval mean was biased 0.013 too high. *****

*AC: The point has been modified as "While both the algorithms did not match the cruise estimate during most of the dust season due to the presence of elevated aerosols, in few cases during ICARB, OMI performed better than OMI-MODIS. OMI performed better due to the better assumption of dust model in the algorithm and (or) wrong model assumption by MODIS. During winter, when the aerosols were present closer to the surface, OMI-MODIS was a bit more consistent compared to OMI. This may be due to scenarios where the CALIPSO climatology is absent and OMI uses its previous aerosol model assumptions. This can also be due to uncertainties in ALH value even after the improvement in the OMI algorithm with the addition of CALIPSO climatology."*

RC: Lines 471-472. I'm not sure about this point at all.

*AC: The corresponding point has been removed.*

RC: Line 474-475. ****What makes you say that the OMI-MODIS is able to detect absorbing aerosols much better than OMI? Detect is not the same as retrieving SSA. Keep that in mind. Note also in the global maps OMI has much better coverage than OMI-MODIS. Why? You never discussed that and it's important. Is OMI reporting cloud contaminated results? Or is OMI much better at detection? *****

*AC: The following paragraph has been added "OMI retrieves aerosol properties at high cloud fraction (Gassó and Torres, 2016) implying two things, either OMI can detect aerosols present above clouds or the OMI pixel was prone to cloud contamination. In their study, Gassó and Torres (2016), observed that while MODIS cloud fraction can be used to screen out cloudy pixels in OMI, it cannot be the lone criterion. While they performed for a single case, an analysis of a larger spatial and temporal scale is required. Aerosol type and aerosol layer height play a vital role in the retrieval of aerosol properties. Without the assumption of aerosol type or height, OMI-MODIS can perform SSA retrievals which is consistent with cruise estimates during the winter when the Bay of Bengal is influenced by anthropogenic aerosols present close to the surface. This is not the case when dust aerosols are present. This discrepancy can be attributed to the difference in the aerosol model assumption by MODIS and OMI. This comparison study has very few points for a detailed analysis. Hence, an accurate comparison and validation of such retrieval algorithms can be possible only when there are more ground-based observations available in the UV spectrum on a larger spatial and temporal scale along with vertical profiles of aerosol absorption.*

**Anonymous Referee #2**

**Specific comments:**

RC: A similar study carried out by Gasso and Torres [2016] presents the results on deriving SSA and ALH from OMI-MODIS synergy and discusses the role of cloud contamination in OMI aerosol retrievals, which author misses to explain in greater details. A discussion highlighting important findings of Gasso and Torres [2016] and its (in)consistency with the new results presented in the paper is required.

*AC: We thank the reviewer for the comment. As suggested a discussion on the important findings of Gassó and Torres and its context in the current work has been included in the introduction and the results section.*

RC: Which version of the OMAERUV product does author use in the present study? I assume here that the latest OMAERUV version 1.8.9 has been employed here to derive the results. If not, the author needs to redo the entire analysis with the latest dataset which incorporates a number of changes applied to the previous version of the algorithm.

*AC: The version used initially was version 1.4.2. As suggested by the reviewer, the entire analysis has been redone using the latest OMAERUV version 1.8.9.*

**Introduction:**

RC: A statement or two citing importance of SSA in estimating radiative forcing with appropriate reference would be needed in the introduction. Figure 5 can be used as a reference.

*AC: As suggested by the reviewer, the effect of change in SSA on the estimation of radiative forcing has been added in the introduction along with references. Figure 8 (Earlier Figure 5) has been used to show the importance of SSA through SBDART simulations.*

RC: Line 52-55: Author has completely forgotten to mention about the long-term record of aerosol absorption retrievals from AERONET!

*AC: Table 1 contains the different methods (along with the corresponding references) used by satellites and ground-based measurements to retrieve aerosol absorption (SSA) including AERONET.*

RC: Line 55: Remove "images".

*AC: The word "images" has been removed*

RC: Line 63-65: It is assumed here that the author is referring to the AOD retrievals from satellites. In this case, the statement "they still have a limited success over deserts" is untrue; AOD retrievals over bright surfaces from the near-UV, Deep Blue, and MAIAC algorithms have achieved a great success in retrieving accurate AODs.

*AC: Here SSA retrievals are considered and based on Table 1, it can be seen that over bright surfaces, SSA retrievals have limited success.*

RC: Line 67-69: What do the "large surface reflectance contrasts" means?

*AC: The sentence has been modified as follows "SSA retrieval in UV spectrum hence avoids difficulties encountered over surfaces with high albedo."*

RC: Line 72: The sub-pixel cloud contamination is a result of the larger footprint of size 13 x 24 kmsquare.

*AC: The reason for sub-pixel contamination has been mentioned in the introduction as suggested.*

RC: Line 76: The time difference between the observations from OMI and MODIS is about 7-8 minutes post-2008 period. A brief discussion about the OMI-MODIS combined retrieval approach is needed here.

*AC: As suggested by the reviewer a brief explanation of the OMI-MODIS algorithm has been mentioned in the introduction as follows "The algorithm uses the MODIS AOD as a reference to infer the aerosol layer height and SSA from OMI. This removes any a priori assumption made by the OMI algorithm regarding an aerosol model."*

**Section 2.1 First paragraph**

RC: In addition to the higher sensitivity to aerosol loading and its absorption properties, the 354 and 388 nm wavelengths have negligible interference from trace gases.

*AC: The statement has been modified as follows "The reason behind choosing these wavelengths is the high sensitivity of upwelling radiances to aerosol absorption and the lower influence of surface in measurements due to low reflectance values in the UV region. In addition to this, the wavelengths also have negligible interference from trace gases."*

RC: "sulphate-based" aerosol type was a gross terminology used for the boundary layer aerosols; it should be changed to "background and urban-industrial" aerosol type.

*AC: The corrections have been made as suggested by the reviewer*

RC: Line 111: Rephrase the sentence as "the retrievals are performed reported for the five discrete aerosol layer heights, i.e., surface (exponential profile), 1.5, 3.0, 6.0, and 10.0 km with latter four following a Gaussian distribution. The final set of AOD/SSA/AAOD retrievals is reported at the mean ALH provided by the 30-month long averaged climatology developed using OMI-CALIOP combined observations."

*AC: The sentence has been rephrased. Since the latest version of the product (v1.8.9) is used, a brief explanation regarding the changes in the new version has also been added.*

RC: Line 120: "have helped distinguish carbonaceous aerosols from dust particles"

*AC: The sentence has been rephrased.*

RC: Line 126: "An effective aerosol layer height was calculated using the CALIOP 1064 nm attenuated backscatter weighted by corresponding altitudes"

*AC: The sentence has been modified accordingly.*

RC: Line 128: "..in the OMAERUV retrievals which then validated against the AERONET observations"

*AC: The sentence has been rephrased.*

**Section 3. Algorithm**

RC: MODIS aerosol product reports retrievals at 10 x 10 km spatial resolution at nadir.

*AC: The sentence has been modified.*

RC: Line 160: "..prone to sub-pixel cloud contamination which may result in overestimation in AOD and SSA"

*AC: The sentence has been modified accordingly.*

RC: Line 163: The high accuracy of size-resolved aerosol retrievals with MODIS is because the over-ocean algorithm employs all seven channels (0.47-2.13 micron) in the inversion enabling better characterization of fine and coarse particles.

*AC: The sentence has been rephrased as suggested by the reviewer.*

RC: Line 169: "..constraints the retrievals of AOD and SSA"

*AC: The sentence has been modified.*

RC: Line 179: Does the algorithm use Angstrom Exponent retrieved by MODIS over the ocean?

*AC: The algorithm does not use the Angstrom Exponent retrieved by MODIS. According to Satheesh et al. (2009), the equation used to correct the extrapolated MODIS AOD "corrects for the variable sensitivity of aerosol species to different ranges of the spectra" and is dependent on the amount of fine mode aerosols present. This equation was obtained by plotting the difference in MODIS and AERONET AOD with the difference in AOD at 470nm and 870nm (the difference represents the aerosol spectral curvature)*

**Section 5. Results**

RC: Line 214-215: not a phenomenon, but the instrumental issue. While a brief discussion on row anomaly is provided in Jethva et al. [2014], Torres et al. [2018] discuss it in great details and its effect on the scan dependency on the OMAERUV retrievals.

*AC: The statement has been corrected. Explanation regarding the row anomaly issue along with the references is discussed in Section 2.1.*

**Section 5.1**

RC: The differences could also be attributed to the shape of the dust particle. In the latest OMAERUV (V1.8.9) algorithm, dust is assumed to be of spheroidal shape with axis ratio distribution adopted from Dubovik et al [2006] study. Please refer to Torres et al. [2018] AMT paper; the citation is provided earlier in this report.

*AC: We thank the reviewer for the suggestion. As mentioned, the shape of the dust has also been added as a factor for the difference.*

**Section 6. Summary and conclusion**

RC: In addition to the sub-pixel cloud contamination and ALH, an uncertainty in prescribing surface albedo is another source of error in the retrieval of SSA from space.

*AC: Uncertainty in surface albedo has been added in the list of issues involved in the retrieval of SSA.*

RC: 3. Is OMI unable to retrieve absorbing aerosols for low ALH or does retrievals but under/over-estimate SSA?

*AC: The statement has been corrected as follows "From Fig. 7 it is also seen that OMI overestimated SSA when it overestimated ALH and vice versa. This can be attributed to the wrong assumption of vertical profiles of aerosols."*

RC: 5. Provide the statistics of the cruise vs. satellite comparison.

*AC: The statistics have been added.*

**Figures**

RC: Figure 1. Since the dynamical range of SSA variations in these maps is confined to 0.85-1.0, a narrower color scale covering this range would be desirable.

*AC: A narrower colour scale from 0.85-1.0 has been used to represent global SSA maps.*

RC: Figure 2. For the most part, SSA retrieved from OMI-MODIS synergy is larger than that retrieved from OMI only.

*AC: After using the latest version of OMI, it was seen from Fig. 2 that the majority of OMI-MODIS retrievals was lesser than OMI especially during the JJA and SON seasons*

RC: Figure 4. From where does the author get ALH difference of 8-10 km?

*AC: There were some points over the two study regions, mostly in the open ocean and some near the coast, where the MODIS extrapolated AOD ($AOD_{388}$) values were closer to the OMI algorithm values retrieved at 6km or 10km (based on linear interpolation). However, the final height given by OMI based on the climatology is lesser than ~3.5km. This resulted in the ALH difference to be > 6km. According to Gassó and Torres (2016), this could be due to the difference in MODIS AOD and OMI AOD spectral curvature resulting in unrealistic height estimation. However, this has not been explored in detail in the current work and will have to be looked in the future.*

RC: Figure 8. Add the measures of agreement, i.e., N, RMSD, correlation

*AC: The statistical measures of the agreement have been added. These include the Total number of points, RMSE and the correlation.*

RC: A plot demonstrating the effect of a change in AOD on the change in SSA either through radiative transfer calculations or from satellite data is needed here. The author may choose a representative region here, say the tropical Atlantic Ocean with dust transport from Sahara for such analysis.

*AC: A plot between the change in AOD and change in SSA has been shown in Figure 6b. The following text has been added in the manuscript – "
[revised manuscript text omitted]

---

## Author Response (AR2)

**Author response to referee comments for the manuscript titled "Multi-Satellite Retrieval of SSA using OMI-MODIS algorithm"**

We thank the referee for the valuable comments and suggestions in improving this manuscript. Please note that author comments (AC) are in red font color. The modified manuscript along with the author marked changes are provided after the authors' response.

**Anonymous Referee #2**

**Abstract**:

RC: Line 7: "… and can be a determinant factor in the estimation of aerosol radiative forcing"

*AC: The statement has been modified accordingly.*

RC: Line 22: "cruise-based measurements"

*AC: The statement has been modified accordingly.*

**Introduction**:

RC: Line 29: "certain aerosol types such as carbonaceous aerosols emitted from biomass burning"

*AC: The statement has been modified accordingly.*

RC: Line 86: "…and uncertainty in the assumption of spectral surface albedo"

*AC: The statement has been modified accordingly.*

RC: Line 91: Levy et al. (2013) should be referenced here.

*AC: The reference has been added.*

RC: Line 93-94: The statement "This removes any a priori assumption made by the OMI algorithm regarding aerosol model" is incorrect. Use of MODIS AOD as an input make the UV algorithm free to retrieve ALH along with SSA. The aerosol type, particle size distribution, refractive indices, and surface albedo still need to be assumed, as in the standard OMI algorithm.

*AC: The statement has been modified accordingly.*

**Summary and Conclusion**:

RC: Line 485: "…aerosol properties such as aerosol optical depth.."

*AC: The statement has been modified accordingly.*

RC: Line 486-487: "…aerosol composition, size, and the wavelength…"

*AC: The statement has been modified accordingly.*

RC: Line 490: "However, these retrievals might be affected by the cloud contamination due to the coarser pixel resolution of 13 x 24 km-square…and are sensitive to the assumption of aerosol layer height in the inversion procedure"

*AC: The statement has been modified accordingly.*

RC: Line 495: "…much larger spatial and temporal scales"

*AC: The statement has been modified accordingly.*

RC: Line 497-499: "…large amounts of aerosols with moderate to high absorption capacity"

*AC: The statement has been modified accordingly.*

RC: Line 501: "…over ATL for more than 80% of the time"

*AC: The statement has been modified accordingly.*

RC: Line 503: Discrepancy in SSA over the Arabian Sea could also be partly attributed to the sub-pixel cloud contamination during summer-monsoon season.

*AC: The statement has been added accordingly.*

RC: Line 513: "…due to the presence of elevated aerosols not sampled by surface instrument"

*AC: The statement has been modified accordingly.*

[revised manuscript text omitted]